# SCALING-TRANSLATION-EQUIVARIANT NETWORKS WITH DECOMPOSED CONVOLUTIONAL FILTERS

## ABSTRACT

Encoding the scale information explicitly into the representation learned by a convolutional neural network (CNN) is beneficial for many vision tasks especially when dealing with multiscale inputs. We study, in this paper, a scaling-translation-equivariant ($\mathcal{ST}$-equivariant) CNN with joint convolutions across the space and the scaling group, which is shown to be both sufficient and necessary to achieve $\mathcal{ST}$-equivariant representations. To reduce the model complexity and computational burden, we decompose the convolutional filters under two pre-fixed separable bases and truncate the expansion to low-frequency components. A further benefit of the truncated filter expansion is the improved deformation robustness of the equivariant representation. Numerical experiments demonstrate that the proposed scaling-translation-equivariant networks with decomposed convolutional filters (ScDCFNet) achieves significantly improved performance in multiscale image classification and better interpretability than regular CNNs at a reduced model size.

## 1 INTRODUCTION

Convolutional neural networks (CNNs) have achieved great success in machine learning problems such as image classification (Krizhevsky et al., 2012), object detection (Ren et al., 2015), and semantic segmentation (Long et al., 2015; Ronneberger et al., 2015). Compared to fully-connected networks, CNNs through spatial weight sharing have the benefit of being translation-equivariant, i.e., translating the input leads to a translated version of the output. This property is crucial for many vision tasks such as image recognition and segmentation. However, regular CNNs are not equivariant to other important group transformations such as rescaling and rotation, and it is beneficial in some applications to also encode such group information explicitly into the network representation.

Several network architectures have been designed to achieve (2D) roto-translation-equivariance ($SE(2)$-equivariance) (Cheng et al., 2019; Marcos et al., 2017; Weiler et al., 2018b; Worrall et al., 2017; Zhou et al., 2017), i.e., roughly speaking, if the input is spatially rotated and translated, the output is transformed accordingly. The feature maps of such networks typically include an extra index for the rotation group $SO(2)$. Building on the idea of group convolutions proposed by Cohen & Welling (2016) for discrete symmetry groups, Cheng et al. (2019) and Weiler et al. (2018b) constructed $SE(2)$-equivariant CNNs by conducting group convolutions jointly across the space and $SO(2)$ using steerable filters (Freeman & Adelson, 1991). Scaling-translation-equivariant ($\mathcal{ST}$-equivariant) CNNs, on the other hand, have typically been studied in a less general setting in the existing literature (Kanazawa et al., 2014; Marcos et al., 2018; Xu et al., 2014; Ghosh & Gupta, 2019). In particular, to the best of our knowledge, a joint convolution across the space and the scaling group $\mathcal{S}$ has yet been proposed to achieve equivariance in the most general form. This is possibly because of two difficulties one encounters when dealing with the scaling group: First, unlike $SO(2)$, it is an acyclic and unbounded group; second, an extra index in $\mathcal{S}$ incurs a significant increase in model parameters and computational burden. Moreover, since the scaling transformation is rarely perfect in practice (due to changing view angle or numerical discretization), one needs to quantify and promote the deformation robustness of the equivariant representation (i.e., is the model still "approximately" equivariant if the scaling transformation is "contaminated" by a nuisance input deformation), which, to the best of our knowledge, has yet been studied in prior works.

The purpose of this paper is to address the aforementioned theoretical and practical issues in the construction of $\mathcal{ST}$-equivariant CNN models. Specifically, our contribution is three-fold:

1. We propose a general $\mathcal{ST}$-equivariant CNN architecture with a joint convolution over $\mathbb{R}^2$ and $\mathcal{S}$, which is proved in Section 4 to be both sufficient and necessary to achieve $\mathcal{ST}$-equivariance.
2. A truncated decomposition of the convolutional filters under a pre-fixed separable basis on the two geometric domains ($\mathbb{R}^2$ and $\mathcal{S}$) is used to reduce the model size and computational cost.

3. We prove the representation stability of the proposed architecture up to equivariant scaling action of the input signal.

Our contribution to the family of group-equivariant CNNs is non-trivial; in particular, the scaling group unlike $SO(2)$ is acyclic and non-compact. This poses challenges both in theory and in practice, so that many previous works on group-equivariant CNNs cannot be directly extended. We introduce new algorithm design and mathematical techniques to obtain the first general $\mathcal{ST}$-equivariant CNN in literature with both computational efficiency and proved representation stability.

## 2 RELATED WORK

**Mixed-scale and $\mathcal{ST}$-equivariant CNNs.** Incorporating multiscale information into a CNN representation has been studied in many existing works. The Inception net (Szegedy et al., 2015) and its generalizations (Szegedy et al., 2017; 2016; Li et al., 2019) stack filters of different sizes in a single layer to address the multiscale salient features. Dilated convolutions (Pelt & Sethian, 2018; Wang et al., 2018; Yu & Koltun, 2016; Yu et al., 2017), pyramid architectures (Ke et al., 2017; Lin et al., 2017), and multiscale dense networks (Huang et al., 2017) have also been proposed to take into account the multiscale feature information. Although the effectiveness of such models have been empirically demonstrated in various vision tasks, there is still a lack of interpretability of their ability to encode the input scale information. Group-equivariant CNNs, on the other hand, explicitly encode the group information into the network representation. Cohen & Welling (2016) proposed CNNs with group convolutions that are equivariant to several finite discrete symmetry groups. This idea is later generalized in Cohen et al. (2018) and applied mainly to the rotation groups $SO(2)$ and $SO(3)$ (Cheng et al., 2019; Weiler et al., 2018a;b). Although $\mathcal{ST}$-equivariant CNNs have also been proposed in the literature (Kanazawa et al., 2014; Marcos et al., 2018; Xu et al., 2014; Ghosh & Gupta, 2019), they are typically studied in a less general setting. In particular, none of these previous works proposed to conduct joint convolutions over $\mathbb{R}^2 \times \mathcal{S}$ as a necessary and sufficient condition to impose equivariance, for which reason they are thus variants of a special case of our proposed architecture where the convolutional filters in $\mathcal{S}$ are Dirac delta functions (c.f. Remark 1.) The scale-space semi-group correlation proposed in the concurrent work (Worrall & Welling, 2019) bears the most resemblance to our proposed model, however, their approach is only limited to discrete semigroups, whereas our model does not have such restriction.

**Representation stability to input deformations.** Input deformations typically induce noticeable variabilities within object classes, some of which are uninformative for the vision tasks. Models that are stable to input deformations are thus favorable in many applications. The scattering transform (Bruna & Mallat, 2013; Mallat, 2010; 2012) computes translation-invariant representations that are Lipschitz continuous to deformations by cascading predefined wavelet transforms and modulus poolings. A joint convolution over $\mathbb{R}^2 \times SO(2)$ is later adopted in Sifre & Mallat (2013) to build roto-translation scattering with stable rotation/translation-invariant representations. These models, however, use pre-fixed wavelet transforms in the networks, and are thus nonadaptive to the data. DCFNet (Qiu et al., 2018) combines a pre-fixed filter basis and learnable expansion coefficients in a CNN architecture, achieving both data adaptivity and representation stability inherited from the filter regularity. This idea is later extended by Cheng et al. (2019) to produce $SE(2)$-equivariant representations that are Lipschitz continuous in $L^2$ norm to input deformations modulo a global rotation, i.e., the model stays approximately equivariant even if the input rotation is imperfect. To the best of our knowledge, a theoretical analysis of the deformation robustness of a $\mathcal{ST}$-equivariant CNN has yet been studied, and a direct generalization of the result in Cheng et al. (2019) is futile because the feature maps of a $\mathcal{ST}$-equivariant CNN is typically not in $L^2$ (c.f. Remark 2.)

## 3 $\mathcal{ST}$-EQUIVARIANT CNN AND FILTER DECOMPOSITION

Group-equivariance is the property of a mapping $f : X \to Y$ to commute with the group actions on the domain $X$ and codomain $Y$. More specifically, let $G$ be a group, and $D_g$, $T_g$, respectively, be group actions on $X$ and $Y$. A function $f : X \to Y$ is said to be $G$-equivariant if

$$f(D_g x) = T_g(f(x)), \quad \forall\, g \in G,\ x \in X. \tag{1}$$

$G$-invariance is thus a special case of $G$-equivariance where $T_g = \mathrm{Id}_Y$. For learning tasks where the feature $y \in Y$ is known a priori to change equivariantly to a group action $g \in G$ on the input $x \in X$, e.g. image segmentation should be equivariant to translation, it would be beneficial to reduce the hypothesis space to include only $G$-equivaraint models. In this paper, we consider mainly the scaling-translation group $\mathcal{ST} \cong \mathcal{S} \times \mathbb{R}^2 \cong \mathbb{R} \times \mathbb{R}^2$. Given $g = (\beta, v) \in \mathcal{ST}$ and an input image

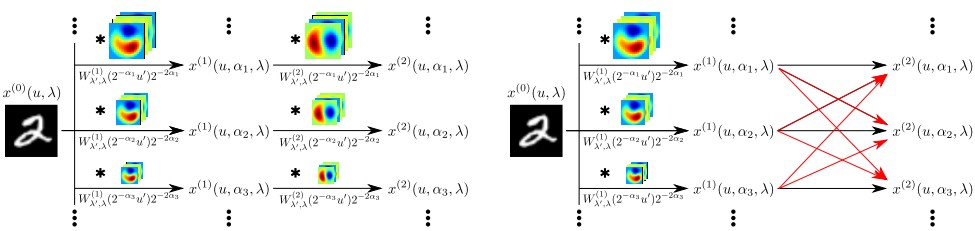

(a) A special case of $\mathcal{ST}$-equivariant CNN with (multiscale) spatial convolutions.

(b) The general case of $\mathcal{ST}$-equivariant CNN with joint convolutions (Theorem 1).

Figure 1: (a) A special case of $\mathcal{ST}$-equivariant CNN with only (multiscale) spatial convolutions. The previous works on $\mathcal{ST}$-equivariant CNNs (Kanazawa et al., 2014; Marcos et al., 2018; Xu et al., 2014; Ghosh & Gupta, 2019) are all variants of this architecture. (b) The general case of $\mathcal{ST}$-equivariant CNN with joint convolutions (Theorem 1) where information transfers among different scales. See Remark 1 for more explanation.

$x^{(0)}(u, \lambda)$ ($u \in \mathbb{R}^2$ is the spatial position, and $\lambda$ is the unstructured channel index, e.g. RGB channels of a color image), the scaling-translation group action $D_g = D_{\beta,v}$ on $x^{(0)}$ is defined as

$$D_{\beta,v} x^{(0)}(u, \lambda) := x^{(0)} \left( 2^{-\beta}(u - v), \lambda \right). \tag{2}$$

Constructing $\mathcal{ST}$-equivariant CNNs thus amounts to finding an architecture $\mathcal{A}$ such that each trained network $f \in \mathcal{A}$ commutes with the group action $D_{\beta,v}$ on the input and a similarly defined group action $T_{\beta,v}$ (to be explained in Section 3.1) on the output.

### 3.1 $\mathcal{ST}$-EQUIVARIANT CNNS

Inspired by Cheng et al. (2019) and Weiler et al. (2018b), we consider $\mathcal{ST}$-equivariant CNNs with an extra index $\alpha \in \mathcal{S}$ for the the scaling group $\mathcal{S} \cong \mathbb{R}$: for each $l \geq 1$, the $l$-th layer output is denoted as $x^{(l)}(u, \alpha, \lambda)$, where $u \in \mathbb{R}^2$ is the spatial position, $\alpha \in \mathcal{S}$ is the scale index, and $\lambda \in [M_l] := \{1, \ldots, M_l\}$ corresponds to the unstructured channels. We use the continuous model for formal derivation, i.e., the images and feature maps have continuous spatial and scale indices. In practice, the images are discretized on a Cartesian grid, and the scales are computed only on a discretized finite interval. Similar to Cheng et al. (2019), the group action $T_{\beta,v}$ on the $l$-th layer output is defined as a scaling-translation in space as well as a shift in the scale channel:

$$T_{\beta,v} x^{(l)}(u, \alpha, \lambda) := x^{(l)} \left( 2^{-\beta}(u - v), \alpha - \beta, \lambda \right), \quad \forall \, l \geq 1. \tag{3}$$

A feedforward neural network is said to be scaling-translation-equivariant, i.e., equivariant to $\mathcal{ST}$, if

$$x^{(l)}[D_{\beta,v} x^{(0)}] = T_{\beta,v} x^{(l)}[x^{(0)}], \quad \forall \, l \geq 1, \tag{4}$$

where we slightly abuse the notation $x^{(l)}[x^{(0)}]$ to denote the $l$-th layer output given the input $x^{(0)}$. The following Theorem shows that $\mathcal{ST}$-equivariance is achieved if and only if joint convolutions are conducted over $\mathcal{S} \times \mathbb{R}^2$ as in (5) and (6).

**Theorem 1.** *A feedforward neural network with an extra index $\alpha \in \mathcal{S}$ for layerwise output is $\mathcal{ST}$-equivariant if and only if the layerwise operations are defined as (5) and (6):*

$$x^{(1)}[x^{(0)}](u, \alpha, \lambda) = \sigma \left( \sum_{\lambda'} \int_{\mathbb{R}^2} 2^{-2\alpha} x^{(0)}(u + u', \lambda') W_{\lambda', \lambda}^{(1)} \left( 2^{-\alpha} u' \right) du' + b^{(1)}(\lambda) \right), \tag{5}$$

$$x^{(l)}[x^{(l-1)}](u, \alpha, \lambda) = \sigma \left( \sum_{\lambda'} \int_{\mathbb{R}^2} \int_{\mathbb{R}} 2^{-2\alpha} x^{(l-1)}(u + u', \alpha + \alpha', \lambda') \cdot \right.$$
$$\left. W_{\lambda', \lambda}^{(l)} \left( 2^{-\alpha} u', \alpha' \right) d\alpha' du' + b^{(l)}(\lambda) \right), \quad \forall l > 1, \tag{6}$$

*where $\sigma : \mathbb{R} \to \mathbb{R}$ is a pointwise nonlinear function.*

We defer the proof of Theorem 1, as well as those of other theorems, to the appendix. We note that the joint-convolution in Theorem 1 is a generalization of the group convolution proposed by Cohen & Welling (2016) to a non-compact group $\mathcal{ST}$ in the continuous setting.

**Remark 1.** *When the (joint) convolutional filter $W_{\lambda',\lambda}^{(l)}(u,\alpha)$ takes the special form $W_{\lambda',\lambda}^{(l)}(u,\alpha) = V_{\lambda',\lambda}^{(l)}(u)\delta(\alpha)$, the joint convolution (6) over $\mathbb{R}^2 \times \mathcal{S}$ reduces to only a (multiscale) spatial convolution*

$$x^{(l)}[x^{(l-1)}](u,\alpha,\lambda) = \sigma\left(\sum_{\lambda'}\int_{\mathbb{R}^2} x^{(l-1)}(u+u',\alpha,\lambda')V_{\lambda',\lambda}^{(l)}\left(2^{-\alpha}u'\right)2^{-2\alpha}du' + b^{(l)}(\lambda)\right), \quad (7)$$

*i.e., the feature maps at different scales do not transfer information among each other (see Figure 1a). The previous works (Kanazawa et al., 2014; Marcos et al., 2018; Xu et al., 2014; Ghosh & Gupta, 2019) on $\mathcal{ST}$-equivariant CNNs are all based on this special case of Theorem 1.*

Although the joint convolutions (6) on $\mathbb{R}^2 \times \mathcal{S}$ provide the most general way of imposing $\mathcal{ST}$-equivariance, they unfortunately also incur a significant increase in the model size and computational burden. Following the idea of Cheng et al. (2019) and Qiu et al. (2018), we address this issue by taking a truncated decomposition of the convolutional filters under a pre-fixed separable basis, which will be discussed in detail in the next section.

## 3.2 SEPARABLE BASIS DECOMPOSITION

We consider decomposing the convolutional filters $W_{\lambda',\lambda}^{(l)}(u,\alpha)$ under the product of two function bases, $\{\psi_k(u)\}_k$ and $\{\varphi_m(\alpha)\}_m$, which are the eigenfunctions of the Dirichlet Laplacian on, respectively, the unit disk $D \subset \mathbb{R}^2$ and $[-1,1]$, i.e.,

$$\begin{cases} \Delta\psi_k = -\mu_k\psi_k & \text{in } D, \\ \psi_k = 0 & \text{on } \partial D, \end{cases} \quad \text{and} \quad \begin{cases} \varphi_m'' = -\nu_m\varphi_m & \text{in } [-1,1] \\ \varphi_m(-1) = \varphi_m(1) = 0. \end{cases} \quad (8)$$

In particular, the spatial basis $\{\psi_k\}_k$ satisfying (8) is the Fourier-Bessel (FB) basis (Abramowitz & Stegun, 1965). In the continuous formulation, the spatial "pooling" operation is equivalent to rescaling the convolutional filters in space. We thus assume, without loss of generality, that the convolutional filters are compactly supported as follows

$$W_{\lambda',\lambda}^{(1)} \in C_c(2^{j_1}D), \quad \text{and} \quad W_{\lambda',\lambda}^{(l)} \in C_c(2^{j_l}D \times [-1,1]), \quad \forall\, l > 1. \quad (9)$$

Let $\psi_{j,k}(u) := 2^{-2j}\psi_k(2^{-j}u)$, then $W_{\lambda',\lambda}^{(l)}$ can be decomposed under $\{\psi_{j_l,k}\}_k$ and $\{\varphi_m\}_m$ as

$$W_{\lambda',\lambda}^{(1)}(u) = \sum_k a_{\lambda',\lambda}^{(1)}(k)\psi_{j_1,k}(u), \; W_{\lambda',\lambda}^{(l)}(u,\alpha) = \sum_m\sum_k a_{\lambda',\lambda}^{(l)}(k,m)\psi_{j_l,k}(u)\varphi_m(\alpha), \; l > 1 \quad (10)$$

where $a_{\lambda',\lambda}^{(1)}(k)$ and $a_{\lambda',\lambda}^{(l)}(k,m)$ are the expansion coefficients of the filters. During training, the basis functions are fixed, and only the expansion coefficients are updated. In practice, we truncate the expansion to only low-frequency components (i.e., $a_{\lambda',\lambda}^{(l)}(k,m)$ are non-zero only for $k \in [K]$, $m \in [K_\alpha]$), which are kept as the trainable parameters. Similar idea has also been considered in the prior works (Qiu et al., 2018; Cheng et al., 2019; Jacobsen et al., 2016). This directly leads to a reduction of network parameters and computational burden. More specifically, let us compare the $l$-th convolutional layer (6) of a $\mathcal{ST}$-equivariant CNN with and without truncated basis decomposition:

**Number of trainable parameters:** Suppose the filters $W_{\lambda',\lambda}^{(l)}(u,\alpha)$ are discretized on a Cartesian grid of size $L \times L \times L_\alpha$. The number of trainable parameters at the $l$-th layer of a $\mathcal{ST}$-equivariant CNN without basis decomposition is $L^2 L_\alpha M_{l-1}M_l$. On the other hand, in an ScDCFNet with truncated basis expansion up to $K$ leading coefficients for $u$ and $K_\alpha$ coefficients for $\alpha$, the number of parameters is instead $KK_\alpha M_{l-1}M_l$. Hence a reduction to a factor of $KK_\alpha/L^2 L_\alpha$ in trainable parameters is achieved for ScDCFNet via truncated basis decomposition. In particular, if $L = 5, L_\alpha = 5, K = 8$, and $K_\alpha = 3$, then the number of parameters is reduced to $(8 \times 3)/(5^2 \times 5) = 19.2\%$.

**Computational cost:** Suppose the size of the input $x^{(l-1)}(u,\alpha,\lambda)$ and output $x^{(l)}(u,\alpha,\lambda)$ at the $l$-th layer are, respectively, $W \times W \times N_\alpha \times M_{l-1}$ and $W \times W \times N_\alpha \times M_l$, where $W \times W$ is the spatial dimension, $N_\alpha$ is the number of scale channels, and $M_{l-1}(M_l)$ is the number of the unstructured input (output) channels. Let the filters $W_{\lambda',\lambda}^{(l)}(u,\alpha)$ be discretized on a Cartesian grid of size $L \times L \times L_\alpha$. The following theorem shows that, compared to a regular $\mathcal{ST}$-equivariant CNN, the computational cost in a forward pass of ScDCFNet is reduced again to a factor of $KK_\alpha/L^2 L_\alpha$.

**Theorem 2.** *Assume $M_l \gg L^2, L_\alpha$, i.e., the number of the output channels is much larger than the size of the convolutional filters in $u$ and $\alpha$, then the computational cost of an ScDCFNet is reduced to a factor of $KK_\alpha/L^2 L_\alpha$ when compared to a $\mathcal{ST}$-equivariant CNN without basis decomposition.*

## 4 REPRESENTATION STABILITY OF SCDCFNET TO INPUT DEFORMATION

Apart from reducing the model size and computational burden, we demonstrate in this section that truncating the filter decomposition has the further benefit of improving the deformation robustness of the equivariant representation, i.e., the equivaraince relation (4) still approximately holds true even if the spatial scaling of the input $D_{\beta,v}x^{(0)}$ is contaminated by a local deformation. The analysis is motivated by the fact that scaling transformations are rarely perfect in practice — they are typically subject to local distortions such as changing view angle or numerical discretization. To quantify the distance between different feature maps at each layer, we define the norm of $x^{(l)}$ as

$$\|x^{(0)}\|^2 = \frac{1}{M_0}\sum_{\lambda=1}^{M_0}\int\left|x^{(0)}(u,\lambda)\right|^2 du, \quad \|x^{(l)}\|^2 = \sup_{\alpha}\frac{1}{M_l}\sum_{\lambda=1}^{M_l}\int\left|x^{(l)}(u,\alpha,\lambda)\right|^2 du, \ l\geq 1. \quad (11)$$

**Remark 2.** *The definition of $\|x^{(l)}\|$ is different from that of RotDCFNet (Cheng et al., 2019), where an $L^2$ norm is taken for the $\alpha$ index as well. The reason why we adopt the $L^\infty$ norm for $\alpha$ in (11) is that $x^{(l)}$ is typically not $L^2$ in $\alpha$, since the scaling group $\mathcal{S}$, unlike $SO(2)$, has infinite Haar measure.*

We next quantify the representation stability of ScDCFNet under three mild assumptions on the convolutional layers and input deformations. First,

**(A1)** The pointwise nonlinear activation $\sigma : \mathbb{R} \to \mathbb{R}$ is non-expansive.

Next, we need a bound on the convolutional filters under certain norms. For each $l \geq 1$, define $A_l$ as

$$\begin{cases} A_1 := \pi\max\left\{\sup_{\lambda}\sum_{\lambda'=1}^{M_0}\|a_{\lambda',\lambda}^{(1)}\|_{\text{FB}}, \ \frac{M_0}{M_1}\sup_{\lambda'}\sum_{\lambda=1}^{M_1}\|a_{\lambda',\lambda}^{(1)}\|_{\text{FB}}\right\}, \\ A_l := \pi\max\left\{\sup_{\lambda}\sum_{\lambda'=1}^{M_{l-1}}\sum_m\|a_{\lambda',\lambda}^{(l)}(\cdot,m)\|_{\text{FB}}, \ \frac{2M_{l-1}}{M_l}\sum_m\sup_{\lambda'}\sum_{\lambda=1}^{M_l}\|a_{\lambda',\lambda}^{(l)}(\cdot,m)\|_{\text{FB}}\right\}, \end{cases} \quad (12)$$

where the Fourier-Bessel (FB) norm $\|a\|_{\text{FB}}$ of a sequence $\{a(k)\}_{k\geq 0}$ is a weighted $l^2$ norm defined as $\|a\|_{\text{FB}}^2 := \sum_k \mu_k a(k)^2$, where $\mu_k$ is the $k$-th eigenvalue of the Dirichlet Laplacian on the unit disk defined in (8). We next assume that each $A_l$ is bounded:

**(A2)** For all $l \geq 1$, $A_l \leq 1$.

The boundedness of $A_l$ is facilitated by truncating the basis decomposition to only low-frequency components (small $\mu_k$), which is one of the key idea of ScDCFNet explained in Section 3.2. After a proper initialization of the trainable coefficients, (A2) can generally be satisfied. The assumption (A2) implies several bounds on the convolutional filters at each layer (c.f. Lemma 2 in the appendix), which, combined with (A1), guarantees that an ScDCFNet is layerwise non-expansive:

**Proposition 1.** *Under the assumption (A1) and (A2), an ScDCFNet satisfies the following.*

*(a) For any $l \geq 1$, the mapping of the $l$-th layer, $x^{(l)}[\cdot]$ defined in (5) and (6), is non-expansive, i.e.,*

$$\|x^{(l)}[x_1] - x^{(l)}[x_2]\| \leq \|x_1 - x_2\|, \quad \forall x_1, x_2. \quad (13)$$

*(b) Let $x_0^{(l)}$ be the $l$-th layer output given a zero bottom-layer input, then $x_0^{(l)}(\lambda)$ depends only on $\lambda$.*

*(c) Let $x_c^{(l)}$ be the centered version of $x^{(l)}$ after removing $x_0^{(l)}$, i.e., $x_c^{(0)}(u,\lambda) := x^{(0)}(u,\lambda) - x_0^{(0)}(\lambda) = x^{(0)}(u,\lambda)$, and $x_c^{(l)}(u,\alpha,\lambda) := x^{(l)}(u,\alpha,\lambda) - x_0^{(l)}(\lambda), \ \forall l \geq 1$, then $\|x_c^{(l)}\| \leq \|x_c^{(l-1)}\|, \ \forall l \geq 1$. As a result, $\|x_c^{(l)}\| \leq \|x_c^{(0)}\| = \|x^{(0)}\|$.*

Finally, we make an assumption on the input deformation modulo a global scale change. Given a $C^2$ function $\tau : \mathbb{R}^2 \to \mathbb{R}^2$, the spatial deformation $D_\tau$ on the feature maps $x^{(l)}$ is defined as

$$D_\tau x^{(0)}(u,\lambda) = x^{(0)}(\rho(u),\lambda), \quad \text{and} \quad D_\tau x^{(l)}(u,\alpha,\lambda) = x^{(l)}(\rho(u),\alpha,\lambda), \quad l \geq 1, \quad (14)$$

where $\rho(u) = u - \tau(u)$. We assume a small local deformation on the input:

**(A3)** $|\nabla\tau|_\infty := \sup_u\|\nabla\tau(u)\| < 1/5$, where $\|\cdot\|$ is the operator norm.

The following theorem demonstrates the representation stability of an ScDCFNet to input deformation modulo a global scale change.

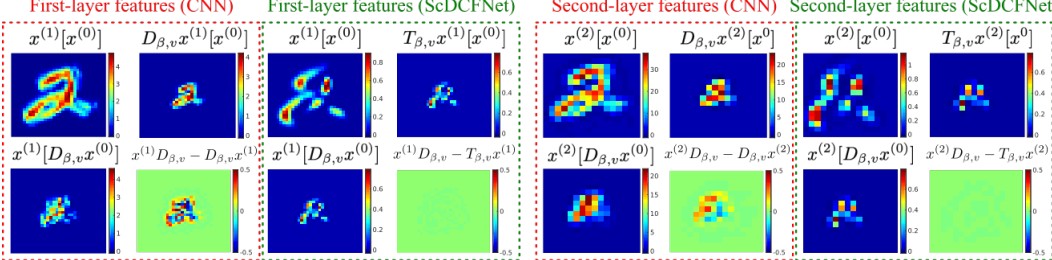

Figure 2: Verification of $\mathcal{ST}$-equivariance in Section 5.1. Given the original input $x^{(0)}$ and its rescaled version $D_{\beta,v}x^{(0)}$, the four figures in each dashed rectangle are: $x^{(l)}[x^{(0)}]$ ($l$-th layer feature of the original input), $x^{(l)}[D_{\beta,v}x^{(0)}]$ ($l$-th layer feature of the rescaled input), $T_{\beta,v}x^{(l)}[x^{(0)}]$ (rescaled $l$-th layer feature of the original input), and the difference $(x^{(l)}[D_{\beta,v}x^{(0)}] - T_{\beta,v}x^{(l)}[x^{(0)}])$ displayed in a (signal intensity) scale relative to the maximum value of $x^{(l)}[D_{\beta,v}x^{(0)}]$. It is clear that even after numerical discretization, $\mathcal{ST}$-equivariance still approximately holds for ScDCFNet, i.e., $x^{(l)}[D_{\beta,v}x^{(0)}] - T_{\beta,v}x^{(l)}[x^{(0)}] \approx 0$, but not for a regular CNN.

**Theorem 3.** *Let $D_\tau$ be a small spatial deformation defined in* (14)*, and let $D_{\beta,v}, T_{\beta,v}$ be the group actions corresponding to an arbitrary scaling $2^{-\beta} \in \mathbb{R}_+$ centered at $v \in \mathbb{R}^2$ defined in* (2) *and* (3)*. In an ScDCFNet satisfying (A1), (A2), and (A3), we have, for any $L$,*

$$\left\| x^{(L)}[D_{\beta,v} \circ D_\tau x^{(0)}] - T_{\beta,v} x^{(L)}[x^{(0)}] \right\| \leq 2^{\beta+1} \left( 4L|\nabla\tau|_\infty + 2^{-j_L}|\tau|_\infty \right) \|x^{(0)}\|. \tag{15}$$

Theorem 3 gauges how approximately equivariant is ScDCFNet if the input undergoes not only a scale change $D_{\beta,v}$ but also a nonlinear spatial deformation $D_\tau$, which is important both in theory and in practice because the scaling of an object is rarely perfect in reality.

## 5 NUMERICAL EXPERIMENTS

In this section, we conduct several numerical experiments for the following three purposes.

1. To verify that ScDCFNet indeed achieves $\mathcal{ST}$-equivariance (4).

2. To illustrate that ScDCFNet significantly outperforms regular CNNs at a much reduced model size in multiscale image classification.

3. To show that a trained ScDCFNet auto-encoder is able to reconstruct rescaled versions of the input by simply applying group actions on the image codes, demonstrating that ScDCFNet indeed explicitly encodes the input scale information into the representation.

The experiments are tested on the Scaled MNIST (SMNIST) and Scaled Fashion-MNIST (SFashion) datasets, which are built by rescaling the original MNIST and Fashion-MNIST (Xiao et al., 2017) images by a factor randomly sampled from a uniform distribution on $[0.3, 1]$. A zero-padding to a size of $28 \times 28$ is conducted after the rescaling. If mentioned explicitly, for some experiments, the images are resized to $64 \times 64$ for better visualization. The implementation details of ScDCFNet are explained in Appendix B.1. In particular, we discuss how to discretize the integral (6) and truncate the scale channel to a finite interval for practical implementation. We also explain how to mitigate the boundary "leakage" effect incurred by the truncation. Moreover, modifications to the spatial pooling and batch-normalization modules of ScDCFNet to maintain $\mathcal{ST}$-equivariance are also explained.

### 5.1 VERIFICATION OF $\mathcal{ST}$-EQUIVARIANCE

We first verify that ScDCFNet indeed achieves $\mathcal{ST}$-equivariance (4). Specifically, we compare the feature maps of a two-layer ScDCFNet with randomly generated truncated filter expansion coefficients and those of a regular CNN. The exact architectures are detailed in Appendix B.2. Figure 2 displays the first- and second-layer feature maps of an original image $x^{(0)}$ and its rescaled version $D_{\beta,v}x^{(0)}$ using the two comparing architectures. Feature maps at different layers are rescaled to the same spatial dimension for visualization. The four images enclosed in each of the dashed rectangle correspond to: $x^{(l)}[x^{(0)}]$ ($l$-th layer feature of the original input), $x^{(l)}[D_{\beta,v}x^{(0)}]$ ($l$-th layer feature of the rescaled input), $T_{\beta,v}x^{(l)}[x^{(0)}]$ (rescaled $l$-th layer feature of the original input, where $T_{\beta,v}$ is understood as $D_{\beta,v}$ for a regular CNN due to the lack of a scale index $\alpha$), and the difference

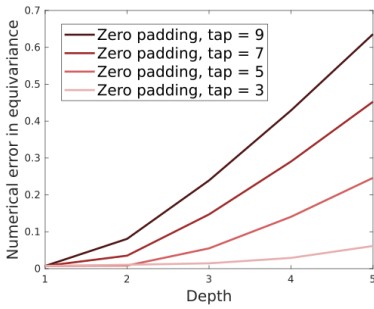 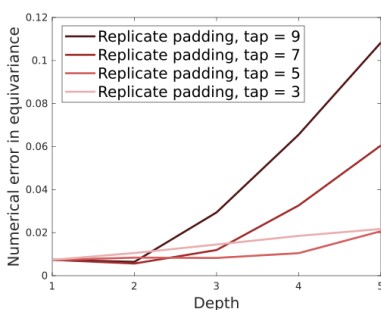

(a) Zero-padding for the scale channel.    (b) Replicate-padding for the scale channel.

Figure 3: The numerical error in equivariance (i.e., the boundary "leakage" effect incurred by scale channel truncation) as a function of network depth. Either (a) zero-padding or (b) replicate-padding is used for the convolution in scale. The error is unavoidable as depth becomes larger, but it can be mitigated by (1) using joint convolutional filters with a smaller support in scale (i.e., a smaller number of "taps" after discretization), and (2) using a replicate-padding instead of zero-padding. See Appendix B.1 for detailed explanation.

$x^{(l)}[D_{\beta,v}x^{(0)}] - T_{\beta,v}x^{(l)}[x^{(0)}]$. It is clear that even with numerical discretization, which can be modeled as a form of input deformation, ScDCFNet is still approximately $\mathcal{ST}$-equivariant, i.e., $x^{(l)}[D_{\beta,v}x^{(0)}] \approx T_{\beta,v}x^{(l)}[x^{(0)}]$, whereas a regular CNN does not have such a property.

We also examin how the numerical error in equivariance (incurred by the the boundary "leakage" effect after the scale channel truncation) evolves as the network gets deeper. The error in equivariance is measured in a relative $L^2$ sense at a particular scale $\alpha$, i.e.,

$$\text{Error} = \|x^{(l)}[D_{\beta,v}x^{(0)}](\cdot,\alpha) - T_{\beta,v}x^{(l)}[x^{(0)}](\cdot,\alpha)\|_{L^2} / \|T_{\beta,v}x^{(l)}[x^{(0)}](\cdot,\alpha)\|_{L^2}. \quad (16)$$

It is clear from Figure 3 that the boundary "leakage" effect is unavoidable as the network becomes deeper. However, the error can be alleviated by either choosing joint convolutional filters with a smaller support in scale (i.e., the filter size, or the number of "taps", after discretization in scale is much smaller compared to the number of scale channels in the feature map), or using a replicate-padding in the scale channel for the joint convolution. See Appendix B.1 for detailed explanation.

## 5.2 MULTISCALE IMAGE CLASSIFICATION

We next demonstrate the improved performance of ScDCFNet in multiscale image classification. The experiments are conducted on SMNIST and SFashion, and a regular CNN is used as a performance benchmark. Both networks are comprised of three convolutional layers with the exact architectures (Table 2) detailed in Appendix B.3. Since the scaling group $\mathcal{S} \cong \mathbb{R}$ is unbounded, we compute only the feature maps $x^{(l)}(u,\alpha,\lambda)$ with the $\alpha$ index restricted to the truncated scale interval $[-1.6, 0]$ ($2^{-1.6} \approx 0.3$), which is discretized uniformly into $N_\alpha = 9$ channels (again, see Appendix B.1 for implementation details.) The performance of the comparing architectures with and without batch-normalization is shown in Table 1. It is clear that, by limiting the hypothesis space to $\mathcal{ST}$-equivaraint models and taking truncated basis decomposition to reduce the model size, ScDCFNet achieves a significant improvement in classification accuracy with a reduced number of trainable parameters. The advantage of ScDCFNet is more pronounced when the number of training samples is small ($N_{\text{tr}} = 2000$), suggesting that, by hardwiring the input scale information directly into its representation, ScDCFNet is less susceptible to overfitting the limited multiscale training data.

We also observe that even when a regular CNN is trained with data augmentation (random cropping and rescaling), its performance is still inferior to that of an ScDCFNet without manipulation of the training data. In particular, although the accuracies of the regular CNNs trained on 2000 SMNIST and SFashion images after data augmentation are improved to, respectively, 93.85% and 79.41%, they still underperform the ScDCFNets without data augmentation (93.91% and 79.94%) using only a fraction of trainable parameters. Moreover, if ScDCFNet is trained with data augmentation, the accuracies can be further improved to 94.30% and 80.62% respectively. This suggests that ScDCFNet can be combined with data augmentation for optimal performance in multiscale image classification.

| Without batch-normalization | | SMNIST test accuracy (%) | | | SFashion test accuracy (%) | |
|---|---|---|---|---|---|---|
| Architectures | Ratio | $N_{tr} = 2000$ | $N_{tr} = 5000$ | $N_{tr} = 10000$ | $N_{tr} = 2000$ | $N_{tr} = 5000$ |
| CNN, $M = 32$ | 1.00 | $92.60 \pm 0.17$ | $94.86 \pm 0.25$ | $96.43 \pm 0.18$ | $77.74 \pm 0.28$ | $82.57 \pm 0.38$ |
| ScDCFNet, $M = 16$ | | | | | | |
| $K = 10, K_\alpha = 3$ | 0.84 | $93.75 \pm 0.02$ | $95.70 \pm 0.09$ | $\mathbf{96.89 \pm 0.10}$ | $78.95 \pm 0.31$ | $\mathbf{83.51 \pm 0.71}$ |
| $K = 8, K_\alpha = 3$ | 0.67 | $\mathbf{93.91 \pm 0.30}$ | $\mathbf{95.71 \pm 0.10}$ | $96.81 \pm 0.12$ | $79.22 \pm 0.50$ | $83.06 \pm 0.32$ |
| $K = 5, K_\alpha = 3$ | 0.42 | $93.52 \pm 0.29$ | $95.19 \pm 0.13$ | $96.77 \pm 0.12$ | $\mathbf{79.74 \pm 0.44}$ | $83.46 \pm 0.69$ |
| $K = 10, K_\alpha = 2$ | 0.56 | $93.68 \pm 0.23$ | $95.54 \pm 0.21$ | $96.87 \pm 0.13$ | $79.01 \pm 0.61$ | $83.43 \pm 0.60$ |
| $K = 8, K_\alpha = 2$ | 0.45 | $93.67 \pm 0.19$ | $95.51 \pm 0.20$ | $96.85 \pm 0.06$ | $79.15 \pm 0.59$ | $83.44 \pm 0.37$ |
| $K = 5, K_\alpha = 2$ | 0.28 | $93.51 \pm 0.30$ | $95.35 \pm 0.21$ | $96.49 \pm 0.17$ | $78.57 \pm 0.53$ | $82.95 \pm 0.46$ |
| ScDCFNet, $M = 8$ | | | | | | |
| $K = 10, K_\alpha = 2$ | 0.14 | $93.68 \pm 0.17$ | $95.21 \pm 0.12$ | $96.51 \pm 0.24$ | $79.11 \pm 0.76$ | $82.92 \pm 0.68$ |
| $K = 8, K_\alpha = 2$ | 0.11 | $93.39 \pm 0.25$ | $95.25 \pm 0.47$ | $96.73 \pm 0.16$ | $78.43 \pm 0.76$ | $82.53 \pm 0.58$ |
| $K = 5, K_\alpha = 2$ | 0.09 | $93.21 \pm 0.20$ | $94.99 \pm 0.12$ | $96.35 \pm 0.12$ | $77.97 \pm 0.37$ | $82.21 \pm 0.67$ |
| $K = 10, K_\alpha = 1$ | 0.09 | $93.26 \pm 0.52$ | $95.22 \pm 0.31$ | $96.52 \pm 0.21$ | $78.71 \pm 0.46$ | $83.05 \pm 0.51$ |
| $K = 8, K_\alpha = 1$ | 0.06 | $93.61 \pm 0.11$ | $95.03 \pm 0.31$ | $96.61 \pm 0.18$ | $78.69 \pm 0.72$ | $82.96 \pm 0.27$ |
| $K = 5, K_\alpha = 1$ | 0.04 | $93.23 \pm 0.13$ | $94.84 \pm 0.38$ | $96.31 \pm 0.17$ | $78.38 \pm 0.81$ | $82.14 \pm 0.25$ |
| With batch-normalization | | SMNIST test accuracy (%) | | | SFashion test accuracy (%) | |
| Architectures | Ratio | $N_{tr} = 2000$ | $N_{tr} = 5000$ | $N_{tr} = 10000$ | $N_{tr} = 2000$ | $N_{tr} = 5000$ |
| CNN, $M = 32$ | 1.00 | $94.78 \pm 0.17$ | $96.58 \pm 0.17$ | $97.41 \pm 0.21$ | $79.79 \pm 0.40$ | $84.38 \pm 0.17$ |
| ScDCFNet, $M = 16$ | | | | | | |
| $K = 10, K_\alpha = 3$ | 0.84 | $95.69 \pm 0.15$ | $96.99 \pm 0.23$ | $97.71 \pm 0.11$ | $81.12 \pm 0.42$ | $84.97 \pm 0.16$ |
| $K = 8, K_\alpha = 3$ | 0.67 | $95.72 \pm 0.29$ | $\mathbf{97.15 \pm 0.24}$ | $\mathbf{97.72 \pm 0.07}$ | $81.41 \pm 0.35$ | $\mathbf{85.11 \pm 0.26}$ |
| $K = 5, K_\alpha = 3$ | 0.42 | $95.31 \pm 0.21$ | $96.75 \pm 0.13$ | $97.38 \pm 0.07$ | $\mathbf{81.73 \pm 0.14}$ | $84.70 \pm 0.19$ |
| $K = 10, K_\alpha = 2$ | 0.56 | $95.58 \pm 0.16$ | $96.92 \pm 0.07$ | $97.61 \pm 0.09$ | $81.21 \pm 0.20$ | $84.77 \pm 0.15$ |
| $K = 8, K_\alpha = 2$ | 0.45 | $\mathbf{95.76 \pm 0.17}$ | $96.74 \pm 0.22$ | $97.68 \pm 0.09$ | $80.84 \pm 0.28$ | $84.70 \pm 0.18$ |
| $K = 5, K_\alpha = 2$ | 0.28 | $95.43 \pm 0.09$ | $96.54 \pm 0.20$ | $97.43 \pm 0.13$ | $81.23 \pm 0.23$ | $84.65 \pm 0.36$ |
| SI-CNN | 1.02 | — | — | $97.25 \pm 0.09$ | — | — |
| VF-SECNN | 1.02 | — | — | $97.56 \pm 0.07$ | — | — |

Table 1: Classification accuracy on the SMNIST and SFashion dataset with and without batch-normalization. The architectures are detailed in Table 2. In particular, $M$ stands for the number of the first-layer (unstructured) output channels, which is doubled after each layer, and $K/K_\alpha$ is the number of basis function in $u/\alpha$ used for filter decomposition. The networks are tested with different training data size, $N_{tr} = 2000, 5000$, and $10000$, and the means and standard deviations after three independent trials are reported. The column "ratio" stands for the ratio between the number of trainable parameters of the current architecture and that of the baseline CNN. The results of the Locally Scale-Invariant CNN (SI-CNN) (Kanazawa et al., 2014) and Vector Field Scale-Equivariant CNN (VF-SECNN) (Marcos et al., 2018) reported in (Marcos et al., 2018) are displayed at the bottom of the table for comparison.

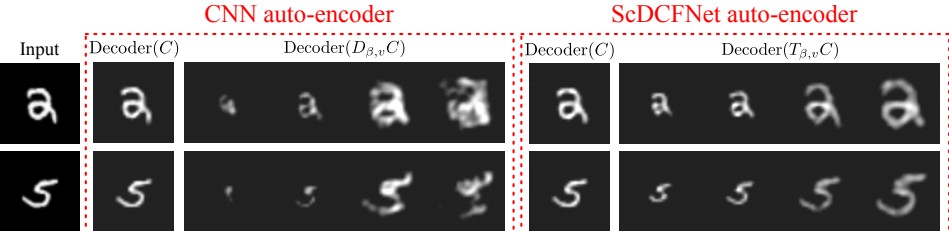

Figure 4: Reconstructing rescaled versions of the original test image by manipulating its image code $C$ according to the group action (3). The first two images on the left are the original inputs; Decoder($C$) denotes the reconstruction using the (unchanged) image code $C$; Decoder($D_{\beta,v}C$) and Decoder($T_{\beta,v}C$) denote the reconstructions using the "rescaled" image codes $D_{\beta,v}C$ and $T_{\beta,v}C$ respectively according to (2) and (3). Unlike the regular CNN auto-encoder, the ScDCFNet auto-encoder manages to generate rescaled versions of the original input, suggesting that it successfully encodes the scale information directly into the representation.

## 5.3 IMAGE RECONSTRUCTION

In the last experiment, we illustrate the ability of ScDCFNet to explicitly encode the input scale information into the representation. To achieve this, we train an ScDCFNet auto-encoder on the SMNIST dataset with images resized to $64 \times 64$ for better visualization. The encoder stacks two $\mathcal{ST}$-equivaraint convolutional blocks with $2 \times 2$ average-pooling, and the decoder contains a succession of two transposed convolutional blocks with $2 \times 2$ upsampling. A regular CNN auto-encoder is also trained for comparison (see Table 3 in Appendix B.4 for the detailed architecture.)

Our goal is to demonstrate that the image code produced by the ScDCFNet auto-encoder contains the scale information of the input, i.e., by applying the group action $T_{\beta,v}$ (3) to the code $C$ of a test

image before feeding it to the decoder, we can reconstruct rescaled versions of original input. This property can be visually verified in Figure 4. In contrast, a regular CNN auto-encoder fails to do so.

## 6 CONCLUSION

We propose, in this paper, a $\mathcal{ST}$-equivaraint CNN with joint convolutions across the space $\mathbb{R}^2$ and the scaling group $\mathcal{S}$, which we show to be both sufficient and necessary to impose $\mathcal{ST}$-equivariant network representation. To reduce the computational cost and model complexity incurred by the joint convolutions, the convolutional filters supported on $\mathbb{R}^2 \times \mathcal{S}$ are decomposed under a separable basis across the two domains and truncated to only low-frequency components. Moreover, the truncated filter expansion leads also to improved deformation robustness of the equivaraint representation, i.e., the model is still approximately equivariant even if the scaling transformation is imperfect. Experimental results suggest that ScDCFNet achieves improved performance in multiscale image classification with greater interpretability and reduced model size compared to regular CNN models.

For future work, we will study the application of ScDCFNet in other more complicated vision tasks, such as object detection/localization and pose estimation, where it is beneficial to directly encode the input scale information into the deep representation. Moreover, the memory usage of our current implementation of ScDCFNet scales linearly to the number of the truncated basis functions in order to realize the reduced computational burden explained in Theorem 2. We will explore other efficient implementation of the model, e.g., using filter-bank type of techniques to compute convolutions with multiscale spatial filters, to significantly reduce both the computational cost and memory usage.

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

## A PROOFS

### A.1 PROOF OF THEOREM 1

*Proof of Theorem 1.* We note first that (4) holds true if and only if the following being valid for all $l \geq 1$,

$$T_{\beta,v} x^{(l)}[x^{(l-1)}] = x^{(l)}[T_{\beta,v} x^{(l-1)}], \tag{17}$$

where $T_{\beta,v} x^{(0)}$ is understood as $D_{\beta,v} x^{(0)}$. We also note that the layer-wise operations of a general feedforward neural network with an extra index $\alpha \in \mathcal{S}$ can be written as

$$x^{(1)}[x^{(0)}](u,\alpha,\lambda) = \sigma\left(\sum_{\lambda'} \int_{\mathbb{R}^2} x^{(0)}(u+u',\lambda') W^{(1)}(u',\lambda',u,\alpha,\lambda) du' + b^{(1)}(\lambda)\right), \tag{18}$$

and, for $l > 1$,

$$x^{(l)}[x^{(l-1)}](u,\alpha,\lambda) = \sigma\left(\sum_{\lambda'} \int_{\mathbb{R}^2} \int_{\mathbb{R}} x^{(l-1)}(u+u',\alpha+\alpha',\lambda')\right.$$
$$\left. W^{(l)}(u',\alpha',\lambda',u,\alpha,\lambda) d\alpha' du' + b^{(l)}(\lambda)\right). \tag{19}$$

To prove the sufficient part: when $l = 1$, (2), (3), and (5) lead to

$$T_{\beta,v} x^{(1)}[x^{(0)}](u,\alpha,\lambda) = x^{(1)}[x^{(0)}]\left(2^{-\beta}(u-v),\alpha-\beta,\lambda\right)$$
$$= \sigma\left(\sum_{\lambda'} \int x^{(0)}\left(2^{-\beta}(u-v)+u',\lambda'\right) W^{(1)}_{\lambda',\lambda}\left(2^{-(\alpha-\beta)}u'\right) 2^{-2(\alpha-\beta)} du' + b^{(1)}(\lambda)\right)$$
$$= \sigma\left(\sum_{\lambda'} \int x^{(0)}\left(2^{-\beta}(u-v+\tilde{u}),\lambda'\right) W^{(1)}_{\lambda',\lambda}\left(2^{-\alpha}\tilde{u}\right) 2^{-2\alpha} d\tilde{u} + b^{(1)}(\lambda)\right), \tag{20}$$

and

$$x^{(1)}[D_{\beta,v} x^{(0)}](u,\alpha,\lambda)$$

$$=\sigma\left(\sum_{\lambda'}\int_{\mathbb{R}^2}D_{\beta,v}x^{(0)}(u+u',\lambda')W^{(1)}_{\lambda',\lambda}\left(2^{-\alpha}u'\right)2^{-2\alpha}du'+b^{(1)}(\lambda)\right)$$

$$=\sigma\left(\sum_{\lambda'}\int_{\mathbb{R}^2}x^{(0)}(2^{-\beta}\left(u+u'-v\right),\lambda')W^{(1)}_{\lambda',\lambda}\left(2^{-\alpha}u'\right)2^{-2\alpha}du'+b^{(1)}(\lambda)\right). \qquad (21)$$

Hence $T_{\beta,v}x^{(1)}[x^{(0)}]=x^{(1)}[D_{\beta,v}x^{(0)}]$.

When $l>1$, we have

$$T_{\beta,v}x^{(l)}[x^{(l-1)}](u,\alpha,\lambda)=x^{(l)}[x^{(l-1)}]\left(2^{-\beta}(u-v),\alpha-\beta,\lambda\right)$$

$$=\sigma\left(\sum_{\lambda'}\int\int x^{(l-1)}\left(2^{-\beta}(u-v)+u',\alpha-\beta+\alpha',\lambda'\right)\cdot\right.$$

$$\left.W^{(l)}_{\lambda',\lambda}\left(2^{-(\alpha-\beta)}u',\alpha'\right)2^{-2(\alpha-\beta)}du'd\alpha'+b^{(l)}(\lambda)\right)$$

$$=\sigma\left(\sum_{\lambda'}\int\int x^{(l-1)}\left(2^{-\beta}(u-v+\tilde{u}),\alpha-\beta+\alpha',\lambda'\right)W^{(l)}_{\lambda',\lambda}\left(2^{-\alpha}\tilde{u},\alpha'\right)2^{-2\alpha}d\tilde{u}d\alpha'+b^{(l)}(\lambda)\right)$$
$$(22)$$

and

$$x^{(l)}[T_{\beta,v}x^{(l-1)}](u,\alpha,\lambda)$$

$$=\sigma\left(\sum_{\lambda'}\int\int T_{\beta,v}x^{(l-1)}(u+u',\alpha+\alpha',\lambda')W^{(l)}_{\lambda',\lambda}\left(2^{-\alpha}u',\alpha'\right)2^{-2\alpha}d\alpha'du'+b^{(l)}(\lambda)\right)$$

$$=\sigma\left(\sum_{\lambda'}\int\int x^{(l-1)}(2^{-\beta}(u+u'-v),\alpha+\alpha'-\beta,\lambda')W^{(l)}_{\lambda',\lambda}\left(2^{-\alpha}u',\alpha'\right)2^{-2\alpha}d\alpha'du'+b^{(l)}(\lambda)\right)$$
$$(23)$$

Therefore $T_{\beta,v}x^{(l)}[x^{(l-1)}]=x^{(l)}[T_{\beta,v}x^{(l-1)}]$, $\forall l>1$.

To prove the necessary part: when $l=1$, we have

$$T_{\beta,v}x^{(1)}[x^{(0)}](u,\alpha,\lambda)=x^{(1)}[x^{(0)}]\left(2^{-\beta}(u-v),\alpha-\beta,\lambda\right)$$

$$=\sigma\left(\sum_{\lambda'}\int x^{(0)}\left(2^{-\beta}(u-v)+u',\lambda'\right)W^{(1)}\left(u',\lambda',2^{-\beta}(u-v),\alpha-\beta,\lambda\right)du'+b^{(1)}(\lambda)\right)$$

$$=\sigma\left(\sum_{\lambda'}\int x^{(0)}\left(2^{-\beta}(u+u'-v),\lambda'\right)\cdot\right.$$

$$\left.W^{(1)}\left(2^{-\beta}u',\lambda',2^{-\beta}(u-v),\alpha-\beta,\lambda\right)2^{-2\beta}du'+b^{(1)}(\lambda)\right), \qquad (24)$$

and

$$x^{(1)}[D_{\beta,v}x^{(0)}](u,\alpha,\lambda)=\sigma\left(\sum_{\lambda'}\int D_{\beta,v}x^{(0)}\left(u+u',\lambda'\right)W^{(1)}\left(u',\lambda',u,\alpha,\lambda\right)du'+b^{(1)}(\lambda)\right)$$

$$=\sigma\left(\sum_{\lambda'}\int x^{(0)}\left(2^{-\beta}(u+u'-v),\lambda'\right)W^{(1)}\left(u',\lambda',u,\alpha,\lambda\right)du'+b^{(1)}(\lambda)\right) \qquad (25)$$

Hence for (17) to hold when $l=1$, we need

$$W^{(1)}\left(u',\lambda',u,\alpha,\lambda\right)=W^{(1)}\left(2^{-\beta}u',\lambda',2^{-\beta}(u-v),\alpha-\beta,\lambda\right)2^{-2\beta},\forall u,\alpha,\lambda,u',\lambda',v,\beta. \quad (26)$$

Keeping $u,\alpha,\lambda,u',\lambda',\beta$ fixed while changing $v$ in (26), we obtain that $W^{(1)}(u',\lambda',u,\alpha,\lambda)$ does not depend on the third variable $u$. Thus $W^{(1)}\left(u',\lambda',u,\alpha,\lambda\right)=W^{(1)}\left(u',\lambda',0,\alpha,\lambda\right)$, $\forall u$. Define $W^{(1)}_{\lambda',\lambda}(u')$ as

$$W^{(1)}_{\lambda',\lambda}(u'):=W^{(1)}\left(u',\lambda',0,0,\lambda\right). \qquad (27)$$

Then, for any given $u', \lambda', u, \alpha, \lambda$, setting $\beta = \alpha$ in (26) leads to

$$W^{(1)}\left(u', \lambda', u, \alpha, \lambda\right) = W^{(1)}\left(2^{-\alpha}u', \lambda', 2^{-\alpha}(u-v), 0, \lambda\right)2^{-2\alpha}$$

$$= W^{(1)}\left(2^{-\alpha}u', \lambda', 0, 0, \lambda\right)2^{-2\alpha} = W^{(1)}_{\lambda',\lambda}\left(2^{-\alpha}u'\right)2^{-2\alpha}. \tag{28}$$

Hence (18) can be written as (5).

For $l > 1$, a similar argument leads to

$$W^{(l)}\left(u', \alpha', \lambda', u, \alpha, \lambda\right) = W^{(l)}\left(2^{-\beta}u', \alpha', \lambda', 2^{-\beta}(u-v), \alpha-\beta, \lambda\right)2^{-2\beta}, \forall u, \alpha, \lambda, u', \alpha', \lambda', v, \beta. \tag{29}$$

Again, keeping $u, \alpha, \lambda, u', \alpha', \lambda', \beta$ fixed while changing $v$ in (29) leads us to the conclusion that $W^{(l)}(u', \alpha', \lambda', u, \alpha, \lambda)$ does not depend on the fourth variable $u$. Define

$$W^{(l)}_{\lambda',\lambda}(u', \alpha') := W^{(l)}\left(u', \alpha', \lambda', 0, 0, \lambda\right). \tag{30}$$

After setting $\beta = \alpha$ in (29), for any given $u', \alpha', \lambda', u, \alpha, \lambda$, we have

$$W^{(l)}\left(u', \alpha', \lambda', u, \alpha, \lambda\right) = W^{(l)}\left(2^{-\alpha}u', \alpha', \lambda', 2^{-\alpha}(u-v), 0, \lambda\right)2^{-2\alpha}$$

$$= W^{(l)}\left(2^{-\alpha}u', \alpha', \lambda', 0, 0, \lambda\right)2^{-2\alpha} = W^{(l)}_{\lambda',\lambda}\left(2^{-\alpha}u'\right)2^{-2\alpha}. \tag{31}$$

This concludes the proof of the Theorem. $\square$

### A.2 PROOF OF THEOREM 2

*Proof of Theorem 2.* In a regular $\mathcal{ST}$-equivariant CNN, the $l$-th convolutional layer (6) is computed as follows:

$$y(u, \alpha, \alpha', \lambda, \lambda') = \int_{\mathbb{R}^2} x^{(l-1)}(u+u', \alpha+\alpha', \lambda')W^{(l)}_{\lambda',\lambda}\left(2^{-\alpha}u', \alpha'\right)2^{-2\alpha}du', \tag{32}$$

$$z(u, \alpha, \lambda, \lambda') = \int_{\mathbb{R}} y(u, \alpha, \alpha', \lambda, \lambda')d\alpha', \tag{33}$$

$$x^{(l)}(u, \alpha, \lambda) = \sigma\left(\sum_{\lambda'=1}^{M_{l-1}} z(u, \alpha, \lambda, \lambda') + b^{(l)}(\lambda)\right). \tag{34}$$

The spatial convolutions in (32) take $2W^2L^2N_\alpha L_\alpha M_l M_{l-1}$ flops (there are $N_\alpha L_\alpha M_l M_{l-1}$ convolutions in $u$, each taking $2W^2L^2$ flops.) The summation over $\alpha'$ in (33) requires $L_\alpha N_\alpha W^2 M_l M_{l-1}$ flops. The summation over $\lambda'$, adding the bias, and applying the nonlinear activation in (34) requires an additional $W^2N_\alpha M_l(2+M_{l-1})$ flops. Thus the total number of floating point computations in a forward pass through the $l$-th layer of a regular $\mathcal{ST}$-equivariant CNN is

$$W^2N_\alpha M_l(2L^2L_\alpha M_{l-1} + L_\alpha M_{l-1} + M_{l-1} + 2). \tag{35}$$

On the other hand, in an ScDCFNet with separable basis truncation up to $KK_\alpha$ leading coefficients, (6) can be computed via the following steps:

$$y(u, \alpha, \lambda', m) = \int_{\mathbb{R}} x^{(l-1)}(u, \alpha+\alpha', \lambda')\varphi_m(\alpha')d\alpha' \tag{36}$$

$$z(u, \alpha, \lambda', k, m) = \int_{\mathbb{R}^2} y(u+u', \alpha, \lambda', m)\psi_{j_l,k}(2^{-\alpha}u')2^{-2\alpha}du' \tag{37}$$

$$x^{(l)}(u, \alpha, \lambda) = \sigma\left(\sum_{m=1}^{K_\alpha}\sum_{k=1}^{K}\sum_{\lambda'=1}^{M_{l-1}} z(u, \alpha, \lambda', k, m)a^{(l)}_{\lambda',\lambda}(k, m) + b^{(l)}(\lambda)\right). \tag{38}$$

The convolutions in $\alpha$ (36) require $2L_\alpha N_\alpha W^2 K_\alpha M_{l-1}$ flops (there are $W^2 K_\alpha M_{l-1}$ convolutions in $\alpha$, each taking $2L_\alpha N_\alpha$ flops.) The spatial convolutions in (37) take $2W^2 L^2 N_\alpha M_{l-1} K_\alpha K$ flops ($N_\alpha M_{l-1} K_\alpha K$ convolutions in $u$, each taking $2W^2 L^2$ flops.) The last step (38) requires an additional $2W^2N_\alpha M_l(1+KK_\alpha M_{l-1})$ flops. Hence the total number of floating point computation for an ScDCFNet is

$$2W^2N_\alpha(KK_\alpha M_{l-1}M_l + M_l + L^2M_{l-1}K_\alpha K + L_\alpha K_\alpha M_{l-1}). \tag{39}$$

In particular, when $M_l \gg L^2, L_\alpha$, the dominating terms in (35) and (39) are, respectively, $2W^2N_\alpha M_l M_{l-1}L^2 L_\alpha$ and $2W^2N_\alpha M_{l-1}M_l KK_\alpha$. Thus the computational cost in an ScDCFNet has been reduced to a factor of $\frac{KK_\alpha}{L^2 L_\alpha}$. $\square$

A.3    PROOF OF PROPOSITION 1

Before proving Proposition 1, we need the following two lemmas.

**Lemma 1.** *Suppose that $\{\psi_k\}_k$ are the FB bases, and $F(u) = \sum_k a(k)\psi_{j,k}(u) = \sum_k a(k)2^{-2j}\psi_k(2^{-j}u)$ is a smooth function on $2^j\overline{B(0,1)}$, then*

$$\int |F(u)|\,du, \ \int |u|\,|\nabla F(u)|\,du, \ 2^j\int |\nabla F(u)|\,du \leq \pi\|a\|_{FB} = \pi\left(\sum_k \mu_k \cdot a(k)^2\right)^{1/2}. \quad (40)$$

This is Lemma 3.5 and Proposition 3.6 in Qiu et al. (2018) after rescaling $u$. Lemma 1 easily leads to the following lemma.

**Lemma 2.** *Let $a_{\lambda',\lambda}^{(l)}(k,m)$ be the coefficients of the filter $W_{\lambda',\lambda}^{(l)}(u,\alpha)$ under the joint bases $\{\psi_k\}_k$ and $\{\varphi_m\}_m$ defined in (10), and define $W_{\lambda',\lambda,m}^{(l)}(u)$ as*

$$W_{\lambda',\lambda,m}(u) := \sum_k a_{\lambda',\lambda}^{(l)}(k,m)\psi_{j_l,k}(u). \quad (41)$$

*We have*

$$B_{\lambda',\lambda}^{(1)}, C_{\lambda',\lambda}^{(1)}, 2^{j_1}D_{\lambda',\lambda}^{(1)} \leq \pi\|a_{\lambda',\lambda}^{(1)}\|_{FB}, \ \ B_{\lambda',\lambda,m}^{(l)}, C_{\lambda',\lambda,m}^{(l)}, 2^{j_l}D_{\lambda',\lambda,m}^{(l)} \leq \pi\|a_{\lambda',\lambda}^{(l)}(\cdot,m)\|_{FB}, \ \forall l > 1, \quad (42)$$

*where*

$$\begin{cases} B_{\lambda',\lambda}^{(1)} := \int \left|W_{\lambda',\lambda}^{(1)}(u)\right|\,du, & B_{\lambda',\lambda,m}^{(l)} := \int \left|W_{\lambda',\lambda,m}^{(l)}(u)\right|\,du, \quad l > 1, \\[2mm] C_{\lambda',\lambda}^{(1)} := \int |u|\left|\nabla_u W_{\lambda',\lambda}^{(1)}(u)\right|\,du, & C_{\lambda',\lambda,m}^{(l)} := \int |u|\left|\nabla_u W_{\lambda',\lambda,m}^{(l)}(u)\right|\,du, \quad l > 1, \\[2mm] D_{\lambda',\lambda}^{(1)} := \int \left|\nabla_u W_{\lambda',\lambda}^{(1)}(u)\right|\,du, & D_{\lambda',\lambda,m}^{(l)} := \int \left|\nabla_u W_{\lambda',\lambda,m}^{(l)}(u)\right|\,du, \quad l > 1. \end{cases} \quad (43)$$

*We thus have*

$$B_l, C_l, 2^{j_l}D_l \leq A_l, \quad (44)$$

*where*

$$B_1 := \max\left\{\sup_\lambda \sum_{\lambda'=1}^{M_0} B_{\lambda',\lambda}^{(1)}, \ \frac{M_0}{M_1}\sup_{\lambda'}\sum_{\lambda=1}^{M_1} B_{\lambda',\lambda}^{(1)}\right\},$$

$$C_1 := \max\left\{\sup_\lambda \sum_{\lambda'=1}^{M_0} C_{\lambda',\lambda}^{(1)}, \ \frac{M_0}{M_1}\sup_{\lambda'}\sum_{\lambda=1}^{M_1} C_{\lambda',\lambda}^{(1)}\right\}, \quad (45)$$

$$D_1 := \max\left\{\sup_\lambda \sum_{\lambda'=1}^{M_0} D_{\lambda',\lambda}^{(1)}, \ \frac{M_0}{M_1}\sup_{\lambda'}\sum_{\lambda=1}^{M_1} D_{\lambda',\lambda}^{(1)}\right\},$$

*and, for $l > 1$,*

$$B_l := \max\left\{\sup_\lambda \sum_{\lambda'=1}^{M_{l-1}}\sum_m B_{\lambda',\lambda,m}^{(l)}, \ \frac{2M_{l-1}}{M_l}\sum_m B_{l,m}\right\}, \quad B_{l,m} := \sup_{\lambda'}\sum_{\lambda=1}^{M_l} B_{\lambda',\lambda,m}^{(l)},$$

$$C_l := \max\left\{\sup_\lambda \sum_{\lambda'=1}^{M_{l-1}}\sum_m C_{\lambda',\lambda,m}^{(l)}, \ \frac{2M_{l-1}}{M_l}\sum_m C_{l,m}\right\}, \quad C_{l,m} := \sup_{\lambda'}\sum_{\lambda=1}^{M_l} C_{\lambda',\lambda,m}^{(l)}, \quad (46)$$

$$D_l := \max\left\{\sup_\lambda \sum_{\lambda'=1}^{M_{l-1}}\sum_m D_{\lambda',\lambda,m}^{(l)}, \ \frac{2M_{l-1}}{M_l}\sum_m D_{l,m}\right\}, \quad D_{l,m} := \sup_{\lambda'}\sum_{\lambda=1}^{M_l} D_{\lambda',\lambda,m}^{(l)}.$$

*In particular, (A2) implies that $B_l, C_l, 2^{j_l}D_l \leq 1$, $\forall l$.*

*Proof of Proposition 1.* To simplify the notation, we omit $(l)$ in $W^{(l)}_{\lambda',\lambda}$, $W^{(l)}_{\lambda',\lambda,m}$, and $b^{(l)}$, and let $M = M_l$, $M' = M_{l-1}$. The proof of (a) for the case $l = 1$ is similar to Proposition 3.1(a) of Qiu et al. (2018) after noticing the fact that

$$\int_{\mathbb{R}^2} \left| W(2^{-\alpha}u) \right| 2^{-2\alpha} du = \int_{\mathbb{R}^2} |W(u)| \, du, \tag{47}$$

and we include it here for completeness. From the definition of $B_1$ in (45), we have

$$\sup_\lambda \sum_{\lambda'} B^{(1)}_{\lambda',\lambda} \le B_1, \quad \text{and} \quad \sup_{\lambda'} \sum_\lambda B^{(1)}_{\lambda',\lambda} \le B_1 \frac{M}{M'}. \tag{48}$$

Thus, given two arbitrary functions $x_1$ and $x_2$, we have

$$\left| \left( x^{(1)}[x_1] - x^{(1)}[x_2] \right)(u, \alpha, \lambda) \right|^2$$

$$= \left| \sigma \left( \sum_{\lambda'} \int x_1(u + u', \lambda') W_{\lambda',\lambda} \left( 2^{-\alpha} u' \right) 2^{-2\alpha} du' + b(\lambda) \right) \right.$$

$$\left. - \sigma \left( \sum_{\lambda'} \int x_2(u + u', \lambda') W_{\lambda',\lambda} \left( 2^{-\alpha} u' \right) 2^{-2\alpha} du' + b(\lambda) \right) \right|^2$$

$$\le \left| \sum_{\lambda'} \int x_1(u + u', \lambda') W_{\lambda',\lambda} \left( 2^{-\alpha} u' \right) 2^{-2\alpha} du' - \sum_{\lambda'} \int x_2(u + u', \lambda') W_{\lambda',\lambda} \left( 2^{-\alpha} u' \right) 2^{-2\alpha} du' \right|^2$$

$$= \left| \sum_{\lambda'} \int (x_1 - x_2)(u + u', \lambda') W_{\lambda',\lambda} \left( 2^{-\alpha} u' \right) 2^{-2\alpha} du' \right|^2$$

$$\le \left( \sum_{\lambda'} \int |(x_1 - x_2)(u + u', \lambda')|^2 \left| W_{\lambda',\lambda}(2^{-\alpha} u') \right| 2^{-2\alpha} du' \right) \left( \sum_{\lambda'} \int \left| W_{\lambda',\lambda}(2^{-\alpha} u') \right| 2^{-2\alpha} du' \right)$$

$$= \left( \sum_{\lambda'} \int |(x_1 - x_2)(u + u', \lambda')|^2 \left| W_{\lambda',\lambda}(2^{-\alpha} u') \right| 2^{-2\alpha} du' \right) \left( \sum_{\lambda'} B^{(1)}_{\lambda',\lambda} \right)$$

$$\le B_1 \sum_{\lambda'} \int |(x_1 - x_2)(\tilde{u}, \lambda')|^2 \left| W_{\lambda',\lambda}(2^{-\alpha}(\tilde{u} - u)) \right| 2^{-2\alpha} d\tilde{u} \tag{49}$$

Therefore, for any $\alpha$,

$$\sum_\lambda \int \left| \left( x^{(1)}[x_1] - x^{(1)}[x_2] \right)(u, \alpha, \lambda) \right|^2 du$$

$$\le \sum_\lambda \int B_1 \sum_{\lambda'} \int |(x_1 - x_2)(\tilde{u}, \lambda')|^2 \left| W_{\lambda',\lambda}(2^{-\alpha}(\tilde{u} - u)) \right| 2^{-2\alpha} d\tilde{u} du$$

$$= B_1 \sum_{\lambda'} \int |(x_1 - x_2)(\tilde{u}, \lambda')|^2 \left( \sum_\lambda \int \left| W_{\lambda',\lambda}(2^{-\alpha}(\tilde{u} - u)) \right| 2^{-2\alpha} du \right) d\tilde{u}$$

$$= B_1 \sum_{\lambda'} \int |(x_1 - x_2)(\tilde{u}, \lambda')|^2 \left( \sum_\lambda B^{(1)}_{\lambda',\lambda} \right) d\tilde{u}$$

$$\le B_1^2 \frac{M}{M'} \sum_{\lambda'} \int |(x_1 - x_2)(\tilde{u}, \lambda')|^2 d\tilde{u}$$

$$= B_1^2 M \|x_1 - x_2\|^2$$

$$\le M \|x_1 - x_2\|^2, \tag{50}$$

where the last inequality makes use of the fact that $B_1 \le A_1 \le 1$ under (A2) (Lemma 2.) Therefore

$$\|x^{(1)}[x_1] - x^{(1)}[x_2]\|^2 = \sup_\alpha \frac{1}{M} \sum_\lambda \int \left| \left( x^{(1)}[x_1] - x^{(1)}[x_2] \right)(u, \alpha, \lambda) \right|^2 du \le \|x_1 - x_2\|^2. \tag{51}$$

This concludes the proof of (a) for the case $l = 1$. To prove the case for any $l > 1$, we first recall from (46) that

$$\sup_\lambda \sum_{\lambda'} \sum_m B^{(l)}_{\lambda',\lambda,m} \le B_l, \quad \text{and} \quad \sum_m B_{l,m} \le B_l \frac{M}{2M'}, \quad \text{where } B_{l,m} = \sup_{\lambda'} \sum_\lambda B^{(l)}_{\lambda',\lambda,m}. \quad (52)$$

Thus, for two arbitrary functions $x_1$ and $x_2$, we have

$$\left| \left( x^{(l)}[x_1] - x^{(l)}[x_2] \right)(u,\alpha,\lambda) \right|^2$$

$$= \left| \sigma \left( \sum_{\lambda'} \int_{\mathbb{R}^2} \int_{\mathbb{R}} x_1(u+u', \alpha+\alpha', \lambda') W_{\lambda',\lambda} \left( 2^{-\alpha} u', \alpha' \right) 2^{-2\alpha} d\alpha' du' + b(\lambda) \right) \right.$$

$$\left. - \sigma \left( \sum_{\lambda'} \int_{\mathbb{R}^2} \int_{\mathbb{R}} x_2(u+u', \alpha+\alpha', \lambda') W_{\lambda',\lambda} \left( 2^{-\alpha} u', \alpha' \right) 2^{-2\alpha} d\alpha' du' + b(\lambda) \right) \right|^2$$

$$\le \left| \sum_{\lambda'} \int_{\mathbb{R}^2} \int_{\mathbb{R}} (x_1 - x_2)(u+u', \alpha+\alpha', \lambda') W_{\lambda',\lambda} \left( 2^{-\alpha} u', \alpha' \right) 2^{-2\alpha} d\alpha' du' \right|^2$$

$$= \left| \sum_{\lambda'} \int_{\mathbb{R}^2} \int_{\mathbb{R}} (x_1 - x_2)(u+u', \alpha+\alpha', \lambda') 2^{-2\alpha} \sum_m W_{\lambda',\lambda,m} \left( 2^{-\alpha} u' \right) \varphi_m(\alpha') d\alpha' du' \right|^2$$

$$= \left| \sum_{\lambda'} \sum_m \int_{\mathbb{R}^2} G_m(u+u', \alpha, \lambda') 2^{-2\alpha} W_{\lambda',\lambda,m}(2^{-\alpha} u') du' \right|^2$$

$$\le \left( \sum_{\lambda'} \sum_m \int_{\mathbb{R}^2} |G_m(u+u', \alpha, \lambda')|^2 \left| W_{\lambda',\lambda,m}(2^{-\alpha} u') \right| 2^{-2\alpha} du' \right) \cdot$$

$$\left( \sum_{\lambda'} \sum_m \int_{\mathbb{R}^2} \left| W_{\lambda',\lambda,m}(2^{-\alpha} u') \right| 2^{-2\alpha} du' \right)$$

$$= \left( \sum_{\lambda'} \sum_m \int_{\mathbb{R}^2} |G_m(\tilde{u}, \alpha, \lambda')|^2 \left| W_{\lambda',\lambda,m}(2^{-\alpha}(\tilde{u}-u)) \right| 2^{-2\alpha} d\tilde{u} \right) \left( \sum_{\lambda'} \sum_m B^{(l)}_{\lambda',\lambda,m} \right)$$

$$\le B_l \sum_{\lambda'} \sum_m \int_{\mathbb{R}^2} |G_m(\tilde{u}, \alpha, \lambda')|^2 \left| W_{\lambda',\lambda,m}(2^{-\alpha}(\tilde{u}-u)) \right| 2^{-2\alpha} d\tilde{u}, \quad (53)$$

where

$$G_m(u, \alpha, \lambda') := \int_{\mathbb{R}} (x_1 - x_2)(u, \alpha+\alpha', \lambda') \varphi_m(\alpha') d\alpha'. \quad (54)$$

We claim (to be proved later in Lemma 3) that

$$M' \|G_m\|^2 = \sup_\alpha \sum_{\lambda'} \int_{\mathbb{R}^2} |G_m(u, \alpha, \lambda')|^2 du \le 2M' \|x_1 - x_2\|^2, \quad \forall m. \quad (55)$$

Thus, for any $\alpha$,

$$\sum_\lambda \int_{\mathbb{R}^2} \left| \left( x^{(l)}[x_1] - x^{(l)}[x_2] \right)(u,\alpha,\lambda) \right|^2 du$$

$$\le \sum_\lambda \int_{\mathbb{R}^2} B_l \sum_{\lambda'} \sum_m \int_{\mathbb{R}^2} |G_m(\tilde{u}, \alpha, \lambda')|^2 \left| W_{\lambda',\lambda,m}(2^{-\alpha}(\tilde{u}-u)) \right| 2^{-2\alpha} d\tilde{u} du$$

$$= B_l \sum_{\lambda'} \sum_m \int_{\mathbb{R}^2} |G_m(\tilde{u}, \alpha, \lambda')|^2 \left( \sum_\lambda \int_{\mathbb{R}^2} \left| W_{\lambda',\lambda,m}(2^{-\alpha}(\tilde{u}-u)) \right| 2^{-2\alpha} du \right) d\tilde{u}$$

$$= B_l \sum_{\lambda'} \sum_m \int_{\mathbb{R}^2} |G_m(\tilde{u}, \alpha, \lambda')|^2 \left( \sum_\lambda B^{(l)}_{\lambda',\lambda,m} \right) d\tilde{u}$$

$$\leq B_l \sum_{\lambda'} \sum_m \int_{\mathbb{R}^2} |G_m(\tilde{u}, \alpha, \lambda')|^2 B_{l,m} d\tilde{u}$$

$$= B_l \sum_m \left( \sum_{\lambda'} \int_{\mathbb{R}^2} |G_m(\tilde{u}, \alpha, \lambda')|^2 d\tilde{u} \right) B_{l,m}$$

$$\leq B_l \cdot 2M' \|x_1 - x_2\|^2 \sum_m B_{l,m}$$

$$\leq B_l^2 \cdot 2M' \|x_1 - x_2\|^2 \frac{M}{2M'} \leq M \|x_1 - x_2\|^2. \tag{56}$$

Therefore

$$\|x^{(l)}[x_1] - x^{(l)}[x_2]\|^2 = \sup_\alpha \frac{1}{M} \sum_\lambda \int_{\mathbb{R}^2} \left| \left( x^{(l)}[x_1] - x^{(l)}[x_2] \right)(u, \alpha, \lambda) \right|^2 du \leq \|x_1 - x_2\|^2. \tag{57}$$

To prove (b), we use the method of induction. When $l = 0$, $x_0^{(0)}(u, \lambda) = 0$ by definition. When $l = 1$, $x_0^{(1)}(u, \alpha, \lambda) = \sigma(b^{(1)}(\lambda))$. Suppose $x_0^{(l-1)}(u, \alpha, \lambda) = x_0^{(l-1)}(\lambda)$ for some $l > 1$, then

$$x_0^{(l)}(u, \alpha, \lambda) = \sigma \left( \sum_{\lambda'} \int_{\mathbb{R}^2} \int_{\mathbb{R}} x_0^{(l-1)}(u + u', \alpha + \alpha', \lambda') W_{\lambda', \lambda}^{(l)} \left( 2^{-\alpha} u', \alpha' \right) 2^{-2\alpha} d\alpha' du' + b^{(l)}(\lambda) \right)$$

$$= \sigma \left( \sum_{\lambda'} x_0^{(l-1)}(\lambda') \int_{\mathbb{R}^2} \int_{\mathbb{R}} W_{\lambda', \lambda}^{(l)} \left( 2^{-\alpha} u', \alpha' \right) 2^{-2\alpha} d\alpha' du' + b^{(l)}(\lambda) \right)$$

$$= \sigma \left( \sum_{\lambda'} x_0^{(l-1)}(\lambda') \int_{\mathbb{R}^2} \int_{\mathbb{R}} W_{\lambda', \lambda}^{(l)} (u', \alpha') d\alpha' du' + b^{(l)}(\lambda) \right)$$

$$= x_0^{(l)}(\lambda). \tag{58}$$

Part (c) is an easy corollary of part (a). More specifically, for any $l > 1$,

$$\|x_c^{(l)}\| = \|x^{(l)} - x_0^{(l)}\| = \|x^{(l)}[x^{(l-1)}] - x_0^{(l)}[x_0^{(l-1)}]\| \leq \|x^{(l-1)} - x_0^{(l-1)}\| = \|x_c^{(l-1)}\|. \tag{59}$$

$\square$

**Lemma 3.** *Suppose $\varphi \in L^2(\mathbb{R})$ with $\mathrm{supp}(\varphi_m) \subset [-1, 1]$ and $\|\varphi\|_{L^2} = 1$, and $x$ is a function of three variables*

$$x : \mathbb{R}^2 \times \mathbb{R} \times [M] \to \mathbb{R} \tag{60}$$

$$(u, \alpha, \lambda) \mapsto x(u, \alpha, \lambda) \tag{61}$$

*with $\|x\|^2 := \sup_\alpha \frac{1}{M} \sum_\lambda \int_{\mathbb{R}^2} |x(u, \alpha, \lambda)|^2 du$. Define $G(u, \alpha, \lambda)$ as*

$$G(u, \alpha, \lambda) := \int_{\mathbb{R}} x(u, \alpha + \alpha', \lambda) \varphi(\alpha') d\alpha. \tag{62}$$

*Then we have*

$$M\|G\|^2 = \sup_\alpha \sum_\lambda \int_{\mathbb{R}^2} |G(u, \alpha, \lambda)|^2 du \leq 2M \|x\|^2. \tag{63}$$

*Proof of Lemma 3.* Notice that, for any $\alpha$, we have

$$\sum_\lambda \int_{\mathbb{R}^2} |G(u, \alpha, \lambda)|^2 du = \sum_\lambda \int_{\mathbb{R}^2} \left| \int_{-1}^{1} x(u, \alpha + \alpha', \lambda) \varphi(\alpha') d\alpha' \right|^2 du$$

$$\leq \sum_\lambda \int_{\mathbb{R}^2} \left( \int_{-1}^{1} |x(u, \alpha + \alpha', \lambda)|^2 d\alpha' \right) \|\varphi\|_{L^2}^2 du$$

$$= \int_{-1}^{1} \left( \sum_{\lambda} \int_{\mathbb{R}^2} |x(u, \alpha + \alpha', \lambda)|^2 \, du \right) d\alpha'$$

$$\leq \int_{-1}^{1} M\|x\|^2 d\alpha' = 2M\|x\|^2. \tag{64}$$

Thus

$$\sup_{\alpha} \sum_{\lambda} \int_{\mathbb{R}^2} |G(u, \alpha, \lambda)|^2 \, du \leq 2M\|x\|^2. \tag{65}$$

$\square$

### A.4 PROOF OF THEOREM 3

To prove Theorem 3, we need the following two Propositions.

**Proposition 2.** *In an ScDCFNet satisfying (A1) and (A3), we have*

1. *For any $l \geq 1$,*

$$\left\| x^{(l)}[D_\tau x^{(l-1)}] - D_\tau x^{(l)}[x^{(l-1)}] \right\| \leq 4(B_l + C_l)|\nabla \tau|_\infty \|x_c^{(l-1)}\|. \tag{66}$$

2. *For any $l \geq 1$, we have*

$$\|T_{\beta,v} x^{(l)}\| = 2^\beta \|x^{(l)}\|, \tag{67}$$

*and*

$$\left\| x^{(l)}[T_{\beta,v} \circ D_\tau x^{(l-1)}] - T_{\beta,v} D_\tau x^{(l)}[x^{(l-1)}] \right\| \leq 2^{\beta+2}(B_l + C_l)|\nabla \tau|_\infty \|x_c^{(l-1)}\|, \tag{68}$$

*where the first $T_{\beta,v}$ in (68) is replaced by $D_{\beta,v}$ when $l = 1$.*

3. *If (A2) also holds true, then*

$$\left\| x^{(l)}[D_{\beta,v} \circ D_\tau x^{(0)}] - T_{\beta,v} D_\tau x^{(l)}[x^{(0)}] \right\| \leq 2^{\beta+3} l |\nabla \tau|_\infty \|x^{(0)}\|, \quad \forall l \geq 1. \tag{69}$$

**Proposition 3.** *In an ScDCFNet satisfying (A1) and (A3), we have, for any $l \geq 1$,*

$$\left\| T_{\beta,v} D_\tau x^{(l)} - T_{\beta,v} x^{(l)} \right\| \leq 2^{\beta+1} |\tau|_\infty D_l \|x_c^{(l-1)}\| \leq 2^{\beta+1} |\tau|_\infty D_l \|x^{(0)}\|. \tag{70}$$

*If (A2) also holds true, then*

$$\left\| T_{\beta,v} D_\tau x^{(l)} - T_{\beta,v} x^{(l)} \right\| \leq 2^{\beta+1-j_l} |\tau|_\infty \|x^{(0)}\|. \tag{71}$$

*Proof of Theorem 3.* Putting together (69) and (71), we have

$$\left\| x^{(L)}[D_{\beta,v} \circ D_\tau x^{(0)}] - T_{\beta,v} x^{(L)}[x^{(0)}] \right\|$$

$$\leq \left\| x^{(L)}[D_{\beta,v} \circ D_\tau x^{(0)}] - T_{\beta,v} D_\tau x^{(L)}[x^{(0)}] \right\| + \left\| T_{\beta,v} D_\tau x^{(L)}[x^{(0)}] - T_{\beta,v} x^{(L)}[x^{(0)}] \right\|$$

$$\leq 2^{\beta+3} L |\nabla \tau|_\infty \|x^{(0)}\| + 2^{\beta+1-j_L} |\tau|_\infty \|x^{(0)}\|$$

$$= 2^{\beta+1} \left( 4L|\nabla \tau|_\infty + 2^{-j_L} |\tau|_\infty \right) \|x^{(0)}\| \tag{72}$$

This concludes the proof of Theorem 3. $\square$

Finally, we need to prove Proposition 2 and Proposition 3, where the following lemma from Qiu et al. (2018) is useful.

**Lemma 4** (Lemma A.1 of Qiu et al. (2018)). *Suppose that $|\nabla \tau|_\infty < 1/5$, $\rho(u) = u - \tau(u)$, then at every point $u \in \mathbb{R}^2$,*

$$||J\rho| - 1| \leq |\nabla \tau|_\infty (2 + |\nabla \tau|_\infty), \tag{73}$$

*where $J\rho$ is the Jacobian of $\rho$, and $|J\rho|$ is the Jacobian determinant. As a result,*

$$||J\rho| - 1|, \left||J\rho^{-1}| - 1\right| \leq 4|\nabla \tau|_\infty, \tag{74}$$

*and,*

$$|J\rho|, \left|J\rho^{-1}\right| \leq 2. \tag{75}$$

*Proof of Proposition 2.* Just like Proposition 1(a), the proof of Proposition 2(a) for the case $l = 1$ is similar to Lemma 3.2 of Qiu et al. (2018) after the change of variable (47). We thus focus only on the proof for the case $l > 1$. To simplify the notation, we denote $x^{(l)}[x^{(l-1)}]$ as $y[x]$, and replace $x_c^{(l-1)}$, $W^{(l)}$, $b^{(l)}$, $M_{l-1}$, and $M_l$, respectively, by $x_c$, $W$, $b$, $M'$, and $M$. By the definition of the deformation $D_\tau$ (14), we have

$$D_\tau y[x](u, \alpha, \lambda) = \sigma \left( \sum_{\lambda'} \int_{\mathbb{R}^2} \int_{\mathbb{R}} x(\rho(u) + u', \alpha + \alpha', \lambda') W_{\lambda', \lambda} \left( 2^{-\alpha} u', \alpha' \right) 2^{-2\alpha} d\alpha' du' + b(\lambda) \right),$$
(76)

$$y[D_\tau x](u, \alpha, \lambda) = \sigma \left( \sum_{\lambda'} \int_{\mathbb{R}^2} \int_{\mathbb{R}} x(\rho(u + u'), \alpha + \alpha', \lambda') W_{\lambda', \lambda} \left( 2^{-\alpha} u', \alpha' \right) 2^{-2\alpha} d\alpha' du' + b(\lambda) \right).$$
(77)

Thus

$$|(D_\tau y[x] - y[D_\tau x])(u, \alpha, \lambda)|^2$$
$$= \left| \sigma \left( \sum_{\lambda'} \int_{\mathbb{R}^2} \int_{\mathbb{R}} x(\rho(u) + u', \alpha + \alpha', \lambda') W_{\lambda', \lambda} \left( 2^{-\alpha} u', \alpha' \right) 2^{-2\alpha} d\alpha' du' + b(\lambda) \right) \right.$$
$$\left. - \sigma \left( \sum_{\lambda'} \int_{\mathbb{R}^2} \int_{\mathbb{R}} x(\rho(u + u'), \alpha + \alpha', \lambda') W_{\lambda', \lambda} \left( 2^{-\alpha} u', \alpha' \right) 2^{-2\alpha} d\alpha' du' + b(\lambda) \right) \right|^2$$
$$\leq \left| \sum_{\lambda'} \int_{\mathbb{R}^2} \int_{\mathbb{R}} \left( x(\rho(u) + u', \alpha + \alpha', \lambda') - x(\rho(u) + u', \alpha + \alpha', \lambda') \right) W_{\lambda', \lambda} \left( 2^{-\alpha} u', \alpha' \right) 2^{-2\alpha} d\alpha' du' \right|^2$$
$$= \left| \sum_{\lambda'} \int_{\mathbb{R}^2} \int_{\mathbb{R}} \left( x_c(\rho(u) + u', \alpha + \alpha', \lambda') - x_c(\rho(u) + u', \alpha + \alpha', \lambda') \right) W_{\lambda', \lambda} \left( 2^{-\alpha} u', \alpha' \right) 2^{-2\alpha} d\alpha' du' \right|^2$$
$$= \left| \sum_{\lambda'} \sum_m \int_{\mathbb{R}^2} \int_{\mathbb{R}} \left( x_c(\rho(u) + u', \alpha + \alpha', \lambda') - x_c(\rho(u) + u', \alpha + \alpha', \lambda') \right) \cdot \right.$$
$$\left. W_{\lambda', \lambda, m} \left( 2^{-\alpha} u' \right) \varphi_m \left( \alpha' \right) 2^{-2\alpha} d\alpha' du' \right|^2,$$
(78)

where the second equality results from the fact that $x(u, \alpha, \lambda) - x_c(u, \alpha, \lambda) = x_0(\lambda)$ depends only on $\lambda$ (Proposition 1(b).) Just like the proof of Proposition 1(a), we take the integral of $\alpha'$ first, and define

$$G_m(u, \alpha, \lambda') \coloneqq \int_{\mathbb{R}} x_c(u, \alpha + \alpha', \lambda') \varphi_m(\alpha') d\alpha'.$$
(79)

Thus

$$|(D_\tau y[x] - y[D_\tau x])(u, \alpha, \lambda)|^2$$
$$\leq \left| \sum_{\lambda'} \sum_m \int_{\mathbb{R}^2} \left( G_m(\rho(u) + u', \alpha, \lambda') - G_m(\rho(u + u'), \alpha, \lambda') \right) W_{\lambda', \lambda, m} \left( 2^{-\alpha} u' \right) 2^{-2\alpha} du' \right|^2$$
$$= \left| \sum_{\lambda'} \sum_m \int_{\mathbb{R}^2} G_m(v, \alpha, \lambda') W_{\lambda', \lambda, m} \left( 2^{-\alpha} (v - \rho(u)) \right) 2^{-2\alpha} dv \right.$$
$$\left. - \sum_{\lambda'} \sum_m \int_{\mathbb{R}^2} G_m(v, \alpha, \lambda') W_{\lambda', \lambda, m} \left( 2^{-\alpha} (\rho^{-1}(v) - u) \right) 2^{-2\alpha} |J\rho^{-1}(v)| dv \right|^2$$
$$= |E_1(u, \alpha, \lambda) + E_2(u, \alpha, \lambda)|^2,$$
(80)

where

$$E_1(u, \alpha, \lambda) = \sum_{\lambda'} \sum_m \int_{\mathbb{R}^2} \left[ W_{\lambda', \lambda, m} \left( 2^{-\alpha} (v - \rho(u)) \right) - W_{\lambda', \lambda, m} \left( 2^{-\alpha} (\rho^{-1}(v) - u) \right) \right] \cdot \quad (81)$$

$$2^{-2\alpha}G_m(v,\alpha,\lambda')dv, \tag{82}$$

$$E_2(u,\alpha,\lambda) = \sum_{\lambda'}\sum_m \int_{\mathbb{R}^2} W_{\lambda',\lambda,m}\left(2^{-\alpha}(\rho^{-1}(v) - u)\right)\left[1 - \left|J\rho^{-1}(v)\right|\right]\cdot \tag{83}$$

$$2^{-2\alpha}G_m(v,\alpha,\lambda')dv. \tag{84}$$

Therefore

$$
\begin{aligned}
M\left\|D_\tau y[x] - y[D_\tau x]\right\|^2 &= \sup_\alpha \sum_\lambda \int_{\mathbb{R}^2} |(D_\tau y[x] - y[D_\tau x])(u,\alpha,\lambda)|^2\, du \\
&\leq \sup_\alpha \sum_\lambda \int_{\mathbb{R}^2} |E_1(u,\alpha,\lambda) + E_2(u,\alpha,\lambda)|^2\, du \\
&= M\|E_1 + E_2\|^2
\end{aligned}
\tag{85}
$$

Hence

$$\left\|D_\tau y[x] - y[D_\tau x]\right\| \leq \|E_1 + E_2\|. \tag{86}$$

We thus seek to estimate $\|E_1\|$ and $\|E_2\|$ individually.

To bound $\|E_2\|$, we let

$$k^{(2)}_{\lambda',\lambda,m}(v,u,\alpha) := W_{\lambda',\lambda,m}\left(2^{-\alpha}(\rho^{-1}(v) - u)\right)\left[1 - \left|J\rho^{-1}(v)\right|\right]2^{-2\alpha}. \tag{87}$$

Then

$$E_2(u,\alpha,\lambda) = \sum_{\lambda'}\sum_m \int_{\mathbb{R}^2} G_m(v,\alpha,\lambda')k^{(2)}_{\lambda',\lambda,m}(v,u,\alpha)dv, \tag{88}$$

and, for any given $v$ and $\alpha$

$$
\begin{aligned}
\int_{\mathbb{R}^2}\left|k^{(2)}_{\lambda',\lambda,m}(v,u,\alpha)\right|du &= \int_{\mathbb{R}^2}\left|W_{\lambda',\lambda,m}\left(2^{-\alpha}(\rho^{-1}(v) - u)\right)\right|\left|1 - \left|J\rho^{-1}(v)\right|\right|2^{-2\alpha}du \\
&= \left|1 - \left|J\rho^{-1}(v)\right|\right|\int_{\mathbb{R}^2}|W_{\lambda',\lambda,m}(\tilde{u})|\,d\tilde{u} \\
&\leq 4|\nabla\tau|_\infty B^{(l)}_{\lambda',\lambda,m},
\end{aligned}
\tag{89}
$$

where the last inequality comes from (74). Moreover, for any given $u$ and $\alpha$,

$$
\begin{aligned}
\int_{\mathbb{R}^2}\left|k^{(2)}_{\lambda',\lambda,m}(v,u,\alpha)\right|dv &= \int_{\mathbb{R}^2}\left|W_{\lambda',\lambda,m}\left(2^{-\alpha}(\rho^{-1}(v) - u)\right)\right|\left|1 - \left|J\rho^{-1}(v)\right|\right|2^{-2\alpha}dv \\
&= \int_{\mathbb{R}^2}\left|W_{\lambda',\lambda,m}(\tilde{v} - 2^{-\alpha}u)\right|\cdot\left||J\rho(2^\alpha\tilde{v})| - 1\right|d\tilde{v} \\
&\leq 4|\nabla\tau|_\infty B^{(l)}_{\lambda',\lambda,m},
\end{aligned}
\tag{90}
$$

where the last inequality is again because of (74). Thus, for any given $\alpha$,

$$
\begin{aligned}
\sum_\lambda \int_{\mathbb{R}^2} |E_2(u,\alpha,\lambda)|^2\, du &= \sum_\lambda \int_{\mathbb{R}^2}\left|\sum_{\lambda'}\sum_m \int_{\mathbb{R}^2} G_m(v,\alpha,\lambda')k^{(2)}_{\lambda',\lambda,m}(v,u,\alpha)dv\right|^2 du \\
&\leq \sum_\lambda \int_{\mathbb{R}^2}\left(\sum_{\lambda'}\sum_m \int_{\mathbb{R}^2} |G_m(v,\alpha,\lambda)|^2\left|k^{(2)}_{\lambda',\lambda,m}(v,u,\alpha)\right|dv\right)\cdot \\
&\qquad\qquad \left(\sum_{\lambda'}\sum_m \int_{\mathbb{R}^2}\left|k^{(2)}_{\lambda',\lambda,m}(v,u,\alpha)\right|dv\right)du \\
&\leq \sum_\lambda \int_{\mathbb{R}^2}\left(\sum_{\lambda'}\sum_m \int_{\mathbb{R}^2} |G_m(v,\alpha,\lambda)|^2\left|k^{(2)}_{\lambda',\lambda,m}(v,u,\alpha)\right|dv\right)\left(\sum_{\lambda'}\sum_m 4|\nabla\tau|_\infty B^{(l)}_{\lambda',\lambda,m}\right)du \\
&\leq 4|\nabla\tau|_\infty B_l \sum_m\sum_{\lambda'} \int_{\mathbb{R}^2} |G_m(v,\alpha,\lambda)|^2\left(\sum_\lambda \int_{\mathbb{R}^2}\left|k^{(2)}_{\lambda',\lambda,m}(v,u,\alpha)\right|du\right)dv
\end{aligned}
$$

$$\leq 4|\nabla\tau|_\infty B_l \sum_m \sum_{\lambda'} \int_{\mathbb{R}^2} |G_m(v,\alpha,\lambda)|^2 \left( \sum_\lambda 4|\nabla\tau|_\infty B^{(l)}_{\lambda',\lambda,m} \right) dv$$

$$\leq 16|\nabla\tau|^2_\infty B_l \sum_m \left( \sum_{\lambda'} \int_{\mathbb{R}^2} |G_m(v,\alpha,\lambda)|^2 \, dv \right) B_{l,m}$$

$$\leq 16|\nabla\tau|^2_\infty B_l \sum_m M'\|G_m\|^2 B_{l,m} \tag{91}$$

Since $\|G_m\|^2 \leq 2\|x_c\|^2$ (by Lemma 3), and $\sum_m B_{l,m} \leq \frac{M}{2M'}B_l$ by definition (46), we thus have

$$\sum_\lambda \int_{\mathbb{R}^2} |E_2(u,\alpha,\lambda)|^2 \, du \leq 16|\nabla\tau|^2_\infty B_l \frac{M}{2M'} B_l \cdot 2M'\|x_c\|^2 = M(4|\nabla\tau|_\infty B_l\|x_c\|)^2, \ \forall\alpha. \tag{92}$$

Taking $\sup_\alpha$ on both sides gives us

$$\|E_2\| \leq 4|\nabla\tau|_\infty B_l\|x_c\|. \tag{93}$$

Similarly, to bound $\|E_1\|$, we introduce

$$k^{(1)}_{\lambda',\lambda,m}(v,u,\alpha) := \left[ W_{\lambda',\lambda,m}\left(2^{-\alpha}(v-\rho(u))\right) - W_{\lambda',\lambda,m}\left(2^{-\alpha}(\rho^{-1}(v)-u)\right) \right] 2^{-2\alpha}. \tag{94}$$

Then

$$E_1(u,\alpha,\lambda) = \sum_{\lambda'} \sum_m \int_{\mathbb{R}^2} G_m(v,\alpha,\lambda') k^{(1)}_{\lambda',\lambda,m}(v,u,\alpha) dv, \tag{95}$$

and, for any given $v$ and $\alpha$, we have

$$\int_{\mathbb{R}^2} \left|k^{(1)}_{\lambda',\lambda,m}(v,u,\alpha)\right| du, \ \int_{\mathbb{R}^2} \left|k^{(1)}_{\lambda',\lambda,m}(v,u,\alpha)\right| dv \leq 4|\nabla\tau|_\infty C^{(l)}_{\lambda',\lambda,m}. \tag{96}$$

The proof of (96) is exactly the same as that of Lemma 3.2 in Qiu et al. (2018) after a change of variable, and we thus omit the detail. Similar to the procedure we take to bound $\|E_2\|$, (96) leads to

$$\|E_1\| \leq 4|\nabla\tau|_\infty C_l\|x_c\|. \tag{97}$$

Putting together (86), (93), and (97), we thus have

$$\|D_\tau y[x] - y[D_\tau x]\| \leq \|E_1 + E_2\| \leq \|E_1\| + \|E_2\| \leq 4(B_l + C_l)|\nabla\tau|_\infty\|x_c\|. \tag{98}$$

This concludes the proof of (a).

To prove (b), given any $\beta \in \mathbb{R}$, and $v \in \mathbb{R}^2$, we have

$$\|T_{\beta,v}x^{(l)}\|^2 = \sup_\alpha \frac{1}{M_l} \sum_\lambda \int_{\mathbb{R}^2} \left| T_{\beta,v}x^{(l)}(u,\alpha,\lambda) \right|^2 du$$

$$= \sup_\alpha \frac{1}{M_l} \sum_\lambda \int_{\mathbb{R}^2} \left| x^{(l)}(2^{-\beta}(u-v),\alpha-\beta,\lambda) \right|^2 du$$

$$= \sup_\alpha \frac{1}{M_l} \sum_\lambda \int_{\mathbb{R}^2} \left| x^{(l)}(\tilde{u},\alpha-\beta,\lambda) \right|^2 2^{2\beta} d\tilde{u}$$

$$= 2^{2\beta}\|x^{(l)}\|^2 \tag{99}$$

Thus (67) holds true. As for (68), we have

$$\left\| x^{(l)}[T_{\beta,v} \circ D_\tau x^{(l-1)}] - T_{\beta,v}D_\tau x^{(l)}[x^{(l-1)}] \right\|$$

$$= \left\| T_{\beta,v}x^{(l)}[D_\tau x^{(l-1)}] - T_{\beta,v}D_\tau x^{(l)}[x^{(l-1)}] \right\|$$

$$= 2^\beta \left\| x^{(l)}[D_\tau x^{(l-1)}] - D_\tau x^{(l)}[x^{(l-1)}] \right\|$$

$$\leq 2^{\beta+2}(B_l + C_l)|\nabla\tau|_\infty\|x^{(l-1)}_c\|, \tag{100}$$

where the first equality holds valid because of Theorem 1, and the second equality comes from (67).

To prove (c), for any $0 \le j \le l$, define $y_j$ as

$$y_j = x^{(l)} \circ x^{(l-1)} \circ \cdots \circ T_{\beta,v} \circ D_\tau x^{(j)} \circ \cdots \circ x^{(0)}. \tag{101}$$

We thus have

$$\left\| x^{(l)}[D_{\beta,v} \circ D_\tau x^{(0)}] - T_{\beta,v} D_\tau x^{(l)}[x^{(0)}] \right\| = \|y_l - y_0\| \le \sum_{j=1}^{l} \|y_j - y_{j-1}\|$$

$$= \sum_{j=1}^{l} \left\| x^{(l)} \circ \cdots \circ T_{\beta,v} \circ D_\tau x^{(j)} \circ \cdots \circ x^{(0)} - x^{(l)} \circ \cdots \circ x^{(j)} \circ T_{\beta,v} \circ D_\tau x^{(j-1)} \circ \cdots \circ x^{(0)} \right\|$$

$$\le \sum_{j=1}^{l} \left\| T_{\beta,v} \circ D_\tau x^{(j)}[x^{(j-1)}] - x^{(j)}[T_{\beta,v} \circ D_\tau x^{(j-1)}] \right\|$$

$$\le \sum_{j=1}^{l} 2^{\beta+2}(B_j + C_j)|\nabla\tau|_\infty \|x_c^{(j-1)}\|$$

$$\le \sum_{k=1}^{l} 2^{\beta+2} \cdot 2|\nabla\tau|_\infty \|x^{(0)}\| = 2^{\beta+3} l |\nabla\tau|_\infty \|x^{(0)}\|, \tag{102}$$

where the second inequality is because of Proposition 1(a), the third inequality is due to (68), and the last inequality holds true because $B_l, C_l \le A_l \le 1$ under (A2) (Lemma 2.) This concludes the proof of Proposition 2. □

*Proof of Proposition 3.* The second inequality in (70) is due to Proposition 1(c). Because of (67), the first inequality in (70) is equivalent to

$$\left\| D_\tau x^{(l)} - x^{(l)} \right\| \le 2|\tau|_\infty D_l \|x_c^{(l-1)}\| \tag{103}$$

Just like Proposition 2(a), the proof of (103) for the case $l = 1$ is similar to Proposition 3.4 of Qiu et al. (2018) after the change of variable (47). A similar strategy as that of Proposition 2(a) can be used to extend the proof to the case $l > 1$. More specifically, denote $x^{(l-1)}, x_c^{(l-1)}, W^{(l)}, b^{(l)}$, respectively, as $x, x_c, W$, and $b$ to simplify the notation. We have

$$\left| \left( D_\tau x^{(l)}[x] - x^{(l)}[x] \right)(u, \alpha, \lambda) \right|^2$$

$$= \left| \sigma \left( \sum_{\lambda'} \int_{\mathbb{R}^2} \int_{\mathbb{R}} x\left(\rho(u) + u', \alpha + \alpha', \lambda'\right) W_{\lambda',\lambda}\left(2^{-\alpha}u', \alpha'\right) 2^{-2\alpha} du' d\alpha' + b(\lambda) \right) \right.$$

$$\left. - \sigma \left( \sum_{\lambda'} \int_{\mathbb{R}^2} \int_{\mathbb{R}} x\left(u + u', \alpha + \alpha', \lambda'\right) W_{\lambda',\lambda}\left(2^{-\alpha}u', \alpha'\right) 2^{-2\alpha} du' d\alpha' + b(\lambda) \right) \right|$$

$$\le \left| \sum_{\lambda'} \int_{\mathbb{R}^2} \int_{\mathbb{R}} x\left(\rho(u) + u', \alpha + \alpha', \lambda'\right) W_{\lambda',\lambda}\left(2^{-\alpha}u', \alpha'\right) 2^{-2\alpha} du' d\alpha' \right.$$

$$\left. - \sum_{\lambda'} \int_{\mathbb{R}^2} \int_{\mathbb{R}} x\left(u + u', \alpha + \alpha', \lambda'\right) W_{\lambda',\lambda}\left(2^{-\alpha}u', \alpha'\right) 2^{-2\alpha} du' d\alpha' \right|$$

$$= \left| \sum_{\lambda'} \int_{\mathbb{R}^2} \int_{\mathbb{R}} x_c\left(\rho(u) + u', \alpha + \alpha', \lambda'\right) \sum_m W_{\lambda',\lambda,m}\left(2^{-\alpha}u'\right) \varphi_m(\alpha') 2^{-2\alpha} du' d\alpha' \right.$$

$$\left. - \sum_{\lambda'} \int_{\mathbb{R}^2} \int_{\mathbb{R}} x_c\left(u + u', \alpha + \alpha', \lambda'\right) \sum_m W_{\lambda',\lambda,m}\left(2^{-\alpha}u'\right) \varphi_m(\alpha') 2^{-2\alpha} du' d\alpha' \right|$$

$$= \left| \sum_{\lambda'} \sum_m \int_{\mathbb{R}^2} G_m(v, \alpha, \lambda') k_{\lambda',\lambda,m}(v, u, \alpha) du' \right|, \tag{104}$$

where

$$G_m(u, \alpha, \lambda') := \int_{\mathbb{R}} x_c(u, \alpha + \alpha', \lambda') \varphi_m(\alpha') d\alpha', \tag{105}$$

$$k_{\lambda', \lambda, m}(v, u, \alpha) := 2^{-2\alpha} \left[ W_{\lambda', \lambda, m} \left( 2^{-\alpha}(v - \rho(u)) \right) - W_{\lambda', \lambda, m} \left( 2^{-\alpha}(v - u) \right) \right]. \tag{106}$$

Similar to (96), we have the following bound

$$\int_{\mathbb{R}^2} |k_{\lambda', \lambda, m}(v, u, \alpha)| \, du, \quad \int_{\mathbb{R}^2} |k_{\lambda', \lambda, m}(v, u, \alpha)| \, dv \leq 2|\nabla \tau|_\infty D^{(l)}_{\lambda', \lambda, m}. \tag{107}$$

Again, the proof of (107) is the same as that of Proposition 3.4 in Qiu et al. (2018) after a change of variable. The rest of the proof follows from a similar argument as in (92) and (93). □

# B  EXPERIMENTAL DETAILS IN SECTION 5

## B.1  IMPLEMENTATION DETAILS

**Discretization and scale channel truncation.** In practice, the scale channel $\mathcal{S} \cong \mathbb{R}$ needs to be truncated to a finite interval $I \subset \mathcal{S}$ such that the layerwise feature map $x^{(l)}(u, \alpha, \lambda)$ is only computed for $\alpha \in I$. This scale channel truncation unavoidably destroys the global scaling symmetry (similar to the fact that truncating an image to a finite spatial support destroys translation symmetry), which leads to the boundary "leakage" effect after a (joint) convolution in scale. This boundary "leakage" effect can be alleviated through the following two ways: (1) we can choose a joint convolutional filter that is compactly supported on a much smaller scale interval $I_\alpha \ni 0$ compared to the truncated scale interval $I$, i.e., $|I_\alpha| \ll |I|$ (a similar idea has been explored in (Worrall & Welling, 2019)); (2) instead of using a zero-padding (which is typically chosen by default as a spatial padding method before spatial convolution), a "replicate"-padding in the scale channel (i.e., extending the truncated scale channel according to Neumann boundary condition) should be implemented before the convolution in scale. After a scale channel truncation, the layerwise feature maps $x^{(l)}(u, \alpha, \lambda)$ and the separable basis functions $\{\psi_k(u)\}_k$, $\{\varphi_m(\alpha)\}_m$ are discretized both in space and scale on a uniform grid. To avoid the spatial aliasing effect, the number $K$ of the spatial basis functions $\{\psi_k(u)\}_{k \in [K]}$ is not allowed to be too large (typically we set $K \leq 10$.)

$\mathcal{ST}$**-equivariant average pooling**. Given a feature $x^{(l)}(u, \alpha, \lambda)$, the traditional average-pooling in $u$ with the same spatial kernel size across $\alpha$ destroys $\mathcal{ST}$-equivariance (4). To remedy this issue, similar to (Zhang, 2019), we first convolve $x^{(l)}$ with a scale-specific low-pass filter before downsampling the convolved signal on a coarser spatial grid. Specifically, we have $\tilde{x}^{(l)}(u, \alpha, \lambda) = \int_{\mathbb{R}^2} x^{(l)}(Nu + u', \alpha, \lambda) \eta(2^{-\alpha} u') 2^{-2\alpha} du'$, where $\tilde{x}^{(l)}$ is the feature after pooling, $\eta$ is a low-pass filter, e.g., a Gaussian kernel, and $N \in \mathbb{Z}_+$ is the pooling factor. We refer to this as $\mathcal{ST}$-equivariant average-pooling.

$\mathcal{ST}$**-equivariant batch-normalization**. Batch-normalization Ioffe & Szegedy (2015) accelerates network training by reducing layerwise covariate shift, and it has become an integral part in various CNN architectures. With an extra scale index $\alpha$ in the feature map $x^{(l)}(u, \alpha, \lambda)$ of an SDCFNet, we need to include $\alpha$ in the normalization in order not to destroy $\mathcal{ST}$-equivariance, i.e., a batch of features $\{x_n^{(l)}(u, \alpha, \lambda)\}_{n=1}^N$ should be normalized as if it were a collection of 3D data (two dimensions for $u$, and one dimension for $\alpha$.)

## B.2  VERIFICATION OF $\mathcal{ST}$-EQUIVARIANCE

The ScDCFNet used in this experiment has two convolutional layers, each of which is composed of a $\mathcal{ST}$-equivariant convolution (5) or (6), a batch-normalization, and a $2 \times 2$ $\mathcal{ST}$-equivariant average-pooling. The expansion coefficients $a_{\lambda', \lambda}^{(1)}(k)$ and $a_{\lambda', \lambda}^{(2)}(k, m)$ are sampled from a Gaussian distribution and truncated to $K = 8$ and $K_\alpha = 3$ leading coefficients for $u$ and $\alpha$ respectively. Similarly, a regular CNN with two convolutional layers and randomly generated $5 \times 5$ convolutional kernels is used as a baseline for comparison.

## B.3  MULTISCALE IMAGE CLASSIFICATION

In the experiments on multiscale image classification on the SMNIST and SFashion dataaset, the network architectures for the ScDCFNet and the regular CNN are shown in Table 2. Stochastic

gradient descent (SGD) with decreasing learning rate from $10^{-2}(10^{-1})$ to $10^{-4}(10^{-3})$ is used to train all networks without (with) batch-normalization for 160 epochs.

| Layer | (Regular) CNN | ScDCFNet |
|-------|---------------|----------|
| 1 | c3x3x1xM  ReLU  ap2x2 | sc(9)9x9x1xM  ReLU  sap2x2 |
| 2 | c3x3xMx2M  ReLU  ap2x2 | sc(9)9x9x$L_\alpha$xMx2M  ReLU  sap2x2 |
| 3 | c3x3x2Mx4M  ReLU  ap2x2 | sc(9)9x9x$L_\alpha$x2Mx4M  ReLU  sap2x2 |
| 4 | fc64  ReLU  fc10  softmax-loss | fc64  ReLU  fc10  softmax-loss |

Table 2: Network architectures used for the experiments in Section 5.2. **cLxLxM'xM:** a regular convolutional layer with M' input channels, M output channels, and LxL spatial kernels. **sc($N_\alpha$)LxLxM'xM:** the first-layer convolution operation (5) in ScDCFNet, where $N_\alpha$ is the number of the uniform grid points to discretize the scale interval $[-1.6, 0]$, and LxL is the spatial kernel size on the largest scale $\alpha = 0$. **sc($N_\alpha$)LxLx$L_\alpha$xM'xM:** the $l$-th layer ($l > 1$) convolution operation (6) in ScDCFNet, where the extra symbol $L_\alpha$ stands for the filter size in $\alpha$. **apLxL(sapLxL):** the regular ($\mathcal{ST}$-equivariant) LxL average-pooling. **fcM:** a fully connected layer with M output channels. Batch-normalization layers are added to each convolutional layer if adopted during training.

## B.4 IMAGE RECONSTRUCTION

The network architectures for the SDCFNet and regular CNN auto-encoders are shown in Table 3. The filter expansion in the SDCFNet auto-encoder is truncated to $K = 8$ and $K_\alpha = 3$. SGD with decreasing learning rate from $10^{-2}$ to $10^{-4}$ is used to train both networks for 20 epochs.

| Layer | Regular auto-encoder | ScDCF auto-encoder |
|-------|----------------------|--------------------|
| 1 | c7x7x1x8  ReLU  ap2x2 | sc(15)13x13x1x4  ReLU  sap2x2 |
| 2 | c7x7x8x16  ReLU  ap2x2 | sc(15)13x13x3x4x8  ReLU  sap2x2 |
| 3 | fc128  ReLU  fc4096  ReLU | fc128  ReLU  fc4096  ReLU |
| 4 | ct7x7x16x8  ReLU  us2x2 | ct7x7x16x8  ReLU  us2x2 |
| 5 | ct7x7x8x1  ReLU  us2x2 | ct7x7x8x1  ReLU  us2x2 |

Table 3: Architectures of the auto-encoders used for the experiment in Section 5.3. The encoded representation is the output of the second layer. **ctLxLxM'xM:** transposed-convolutional layers with M' input channels, M output channels, and LxL spatial kernels. **us2x2:** 2x2 spatial upsampling. See the caption of Table 2 for the definitions of other symbols. Batch-normalization (not shown in the table) is used after each convolutional layer.

