# OpenReview forum: "Scale-Equivariant Neural Networks with Decomposed Convolutional Filters"
_ICLR.cc/2020/Conference — Reject_

### Official Review · AnonReviewer3 · 2019-10-21
**Official Blind Review #3**

**Rating:** 6

**Review:**

*Paper summary*

The authors propose a CNN architecture, that is theoretically equivariant to isotropic scalings and translations. For this they add an extra scale-dimension to activation tensors, along with the existing two spatial dimensions. In practice they implement this with scale-steerable filters, which are discretised and truncated in both the spatial and scale-dimensions. They also provide a deformation robustness analysis.

*Paper decision*

Thank you for writing a very interesting paper indeed. I have to admit I am somewhat on the fence about this paper. I think it contains many nice ideas, but the experimental section is somewhat lacking in terms of comparisons or insights which I can gain, and the theory has some missing elements too (which I shall discuss below). For that reason, I am recommending a weak reject, but would very easily upgrade this if the authors provide strong rebuttal to my comments below.

*Supporting arguments*

-Experiments: The experiments are quite light, although I must admit that many other works in the area of equivariance are also light on experimentation and if there is enough theory, that is not such a great issue. The main issues I have are 1) the choice of experiments, 2) the comparisons against an insufficient number of baselines, and 3) the ablation studies.
1) The choice of experiments: I think looking at scaled-MNIST is not particularly useful as an experiment nowadays, unless used as a toy experiment. A larger dataset with real-life scale variations would have been better. Furthermore, I’m not sure what the image reconstruction task is supposed to tell the reader, that the ScDCFNet is able to generalise to new scales?
2) There is a lot of concurrent work on multiscale architecture. To name a few:
Multigrid Neural Architectures, Ke et al., 2017
Feature Pyramid Networks for Object Detection, Lin et al., 2017
Multi-Scale Dense Networks for Resource Efficient Image Classification, Huang et al., 2018

Deep Scale-spaces, Worrall and Welling, 2019

I believe these works should at least be cited, but ideally compared against.

3) I would have liked to have seen some numerical results for the verification of scale equivariance. In the first layer the equivariance improbably going to be close to perfect because there is no truncation of the scale-dimension in the network, but after this layer I predict the equivariance error increases due to truncations effects. This would be a similar effect to other works, such as “Deep Scale-spaces” (Worrall and Welling, 2019).

-Theory: I think the theory is very interesting and a meaningful contribution in its own right. The authors treat scale-translation in continuous space as a group action on signals. This motivates the convolution presented in Theorem 1. This is a group convolution as per Cohen and Welling, (2016) modified to continuous space for a non-compact group. (Actually, this should be mentioned in the text as a matter of good scholarship). This is nice, since the group convolution has not been used in for a regular representation of a non-compact group (other than translation) as far as I can tell. What is perhaps not clear for me is how the theory breaks down in practice, since the implementation requires discretisation in space AND scale, which is not discussed much and furthermore, filters are restricted in spatial AND scale dimensions, leading to truncation errors in the equivariance. This last perspective was not discussed, and I feel it rather should be.
The theory goes further into deformation stability, which is a fresh perspective in the equivariance literature, so I am happy for its inclusion. Perhaps more motivation for why you think this is necessary would be warmly welcomed.

*Smaller questions/notes for the authors*

- Technically this is scale-translation equivariance, you even write this in the method section of your paper, why is it not in the title? The reason I mention this Is because there are scale equivariant networks, which are not translation equivariant in the literature, see “Warped Convolutions: Efficient Invariance to Spatial Transformations” (Henriques and Vedaldi, 2019).
- Please make the link between Theorem 1 and the group convolution of Cohen and Welling (2016)
- Last paragraph of page 1: A steerable-in-scale filter does exist, see “Deformable kernels for early vision” (Perona, 1991)
- Please use numbering for all display-mode equations.
- This scheme works perfectly in the continuous-image setting, but how about for discretized images? In that case it cannot be scale-translation equivariant because the scaling-action is no longer part of a group.
- In equations 6 and 7, what is the specific motivation for using the laplacian eigen-decompositions as a basis? Is it for steerability with respect to the scale-translation action? Otherwise, surely any basis will do?
- Remark 2: If you are considering a truncation of the scale-axis, surely you can still use an L2 norm when quantifying the robustness of your representation?
- Bandlimiting of the filters: I would consider citing “Structured Receptive Fields in CNNs”, (Jacobsen et al., 2016)
- Pooling: A useful citation here would be “Making Convolutional Networks Shift-Invariant Again” (Zhang, 2019). They precise low pass filter before pooling.
- Experiments:
—I’m not clear on the reason to include reasons with and without batch normalization. This is quite unconventional
— Did you ever use scale augmentation? What is the effect of training on one scale and then testing on another?

**Experience Assessment:**

I have published in this field for several years.

**Review Assessment: Checking Correctness Of Derivations And Theory:**

I carefully checked the derivations and theory.

**Review Assessment: Checking Correctness Of Experiments:**

I carefully checked the experiments.

**Review Assessment: Thoroughness In Paper Reading:**

I read the paper thoroughly.

---

> ### Author Response · Authors · 2019-11-15
> **Response to R3 (part 1)**
>
> We would like to thank the reviewer for reading our manuscript and providing valuable comments. Please see below our detailed response, and the manuscript has also been updated accordingly.
>
>
> **Major comments**
>
> — scaled-MNIST is OK as a toy experiment, but a larger dataset with real-life scale-variations would be better. Furthermore, what the image reconstruction task is supposed to tell the reader?
>
> The purpose of the image reconstruction experiment is to demonstrate that ScDCFNet is able to generalize to new scales. Furthermore, we want to show that the image code produced by an ScDCFNet encoder is more interpretable, i.e., it contains the input scale information, and manipulating the image code before feeding it to the decoder can produced rescaled versions of the input.
> We agree with the reviewer that experiments on a larger dataset with real-life scale variations would be great. Due to limited resources, the current paper does not have such a result. Beyond MNIST and Fashion MNIST, we have conducted extra experiments on scaled-CIFAR10 dataset, where the improvement in classification accuracy in this experiment is not as pronounced as the SMNIST/SFashion MNIST dataset. We think the reason is that the objects in the CIFAR dataset (unlike MNIST and Fashion MNIST) are almost visually indiscernible after rescaling. In other words, small object images like CIFAR are not proper for the current study.
> Meanwhile, as suggested by R3 and a volunteer reviewer, we did conduct an extra experiment on the SMNIST dataset with the conventional 10k/50k train/test splits and compare the result of the ScDCFNet to other benchmark models. The results have been updated in Table 1. We will keep working on optimizing the implementation of our proposed model and its application with larger datasets with real-life scale-variations.
>
>
> —Citing concurrent works and compare against them.
>
> We would like to thank the reviewer for pointing us to the interesting concurrent works on multiscale network architectures. Citations of these prior works have been added in our revised paper. As mentioned before, comparisons between our proposed model against more baselines have been added in Table 1.
>
>
> — More numerical results for the verification of scale-equivariance. In the first layer the equivariance improbably going to be close to perfect because there is no truncation of the scale-dimension in the network, but after this layer I predict the equivariance error increases due to truncations effects.
>
> We have added another numerical result on the verification of scale-equivariance in Figure 3 as requested by the reviewer. The reviewer is absolutely correct that as the network goes deeper, the equivariance relation is subject to numerical contamination incurred by  (1) a truncation of the global scaling group to a finite interval, and (2) the boundary “leakage” issue caused by the non-degenerate joint-convolution, i.e., when the joint filter is not a delta function in the scale dimension (c.f. Remark 1.) There are two ways to mitigate this issue. First, we can choose a joint convolutional filter that is compactly supported on a smaller scale interval (this has also been pointed out in [1].) Both Figure 3(a) and Figure 3(b) show that convolutional filters with a smaller number of “taps” in scale (i.e., smaller filter size in scale) lead to a smaller numerical error in equivariance as the network depth progresses.. Second, instead of using a zero-padding, a “replicate” padding should be implemented for the scale dimension before the scale-convolution. This argument is verified by comparing Figure 3(b) to Figure 3(a) (noticing the different error scales in these two figures.) The detailed explanation has been added in Appendix B.1 and Section 5.1.
>
>
> — The theory is very interesting and a meaningful contribution in its own right. However a reference to [2] as a motivation for Theorem 1 needs to be mentioned in the text.
>
> We thank the reviewer again for kindly acknowledging the theoretical contribution of our paper. A reference to [2] has been added for Theorem 1.

---

> > ### Author Response · Authors · 2019-11-15
> > **Response to R3 (part 2)**
> >
> > — How does the theory break down in practice? How to implement the discretization both in space AND scale? What is the error in equivariance incurred by discretization and truncation both in space AND scale?
> >
> > A detailed discussion on the model implementation has been added in Appendix B.1 as requested by the reviewer. In particular, the layerwise feature maps and separable basis functions (with closed-form expressions) are discretized both in space and scale on a uniform grid. Discretization in scale does not introduce error in the equivariance relation, and the reason is that the group action (2) on the feature maps is a shift in the scale channel — this might be easier to understand by viewing the convolutional filters in scale as a finite combination of Dirac delta functions. Discretization in space does incur an unavoidable numerical error in equivariance, because the scaling action in space is not well defined (and only approximated) in the discrete setting. This puts a restriction on the number of spatial basis functions “K” we can use to avoid the aliasing effect. The numerical error caused by discretization also motivates us to quantify the deformation robustness of the equivariant representation, because discretization can be viewed as a local deformation by mean value theorem. As for the error caused by truncation in scale, this is again unavoidable (just like translation equivariance is subject to “spatial boundary” issue because images are compactly supported.) We have mentioned two ways (explained again in Appendix B.1) to mitigate the truncation error in our response to a previous comment.
> >
> >
> > — Deformation stability is a fresh perspective in the equivariance literature. More motivation would be warmly welcomed.
> >
> > We have added more clarification on the motivation of the stability analysis in the beginning of Section 4. In essence, scaling transformations are rarely perfect in practice, e.g., the scaling effect of a 2D image capturing a 3D object moving towards/away from the camera is subject to image deformation caused by changing view angles. Stability analysis is thus important to quantify the “brittleness” of the equivariance relation, which is an algebraic property, to data deformation for practical usage.
> >
> >
> > **Smaller comments**
> >
> > — Technically this is scale-translation equivariance.
> >
> > We thank the reviewer for pointing this out. This has been updated throughout the revised paper.
> >
> >
> > — Please make the link between Theorem 1 and the group convolution of Cohen and Welling (2016)
> >
> > A reference to [2] has been added as suggested by the reviewer.
> >
> >
> > — Last paragraph of page 1: A steerable-in-scale filter does exist, see “Deformable kernels for early vision” (Perona, 1991.)
> >
> > This has been updated in the revised paper.
> >
> >
> > — Please use numbering for all display-mode equations.
> >
> > This has been updated in the revised paper.
> >
> >
> > — This scheme works perfectly in the continuous-image setting, but how about for discretized images? In that case it cannot be scale-translation equivariant because the scaling-action is no longer part of a group.
> >
> > We agree with the reviewer that the scaling-translation group action is not well-defined in the discrete setting (i.e., on a lattice), and it is thus viewed as an approximation to the continuous limit. This is also one of the reasons why truncated basis expansion is used to alleviate the aliasing effect. We would like to point out that the rotation group action is also not well-defined in the discrete setting (when the rotation angle is non-trivial.)
> >
> >
> > — In equations 6 and 7, what is the specific motivation for using the Laplacian eigen-decompositions as a basis? Is it for steerability with respect to the scale-translation action? Otherwise, surely any basis will do?
> >
> > The motivation is two-folds. First, for realistic implementation, we need the convolutional filters to be compactly supported in space (preferably in an isotropic way.) Thus an intuitive idea is to find an orthonormal function basis with Dirichlet boundary condition on the 2D disk, i.e., the Laplacian eigenfunctions. Second, Lemma 1 in Appendix A.3 also requires the basis functions to be Dirichlet Laplacian eigenfunctions (albeit not necessarily supported on a disk) so that the three quantities in Lemma 1 can be bounded by the weighted L_2 norm of the expansion coefficients.

---

> > > ### Author Response · Authors · 2019-11-15
> > > **Response to R3 (part 3)**
> > >
> > > — Remark 2: If you are considering a truncation of the scale-axis, surely you can still use an L2 norm when quantifying the robustness of your representation?
> > >
> > > This is unfortunately not the case. The scale-axis truncation is merely a practical (and yet unavoidable) numerical procedure to compute (or keep track of) the features at the truncated scale interval. Even though the features outside the scale interval are not computed, they are still there. For example, if we truncate the scale-axis of the feature maps to the interval [-1, 1], then verifying the scale-equivariance relation between the feature maps of “x(u)” and “y(u)=x(2u)” requires us to compare the scale interval of “x^{(1)}(u, alpha)” at [-1, 1] and that of “y^{(1)}(u, alpha)” at [-2, 0], which is not the same as the originally truncated scale interval [-1, 1]. This is because the group action on the feature map includes a shift in the scale channel as well. Thus the only reasonable way to quantify the representation robustness is to compare the feature maps on the entire scaling group (without any truncation.)
> > >
> > >
> > > — Bandlimiting of the filters: I would consider citing “Structured Receptive Fields in CNNs”, (Jacobsen et al., 2016)
> > > — Pooling: A useful citation here would be “Making Convolutional Networks Shift-Invariant Again” (Zhang, 2019).
> > >
> > > We would like to thank the reviewer for pointing us to these interesting references. They have been added in the revised paper.
> > >
> > >
> > > — I’m not clear on the reason to include reasons with and without batch normalization. This is quite unconventional.
> > >
> > > The reason why we include experiments with and without batch-normalization is that the batch-normalization module needs to be modified for ScDCFNet so that scaling-translation-equivariance still holds true. This has been explained in Appendix B.1 and Section 5 (right before Section 5.1.)
> > >
> > >
> > >
> > > —  Did you ever use scale augmentation?
> > >
> > > The result reported in Table 1 does not use scale augmentation. However, we have indeed compared the result of a regular CNN with augmentation and that of an ScDCFNet with/without augmentation, which has been explained in the last paragraph of Section 5.2.
> > >
> > >
> > >
> > > [1] Daniel Worral and Max Welling, “Deep Scale-spaces: Equivariance Over Scale.” arXiv:1905.11697, 2019.
> > > [2] Taco Cohen and Max Welling, “Group Equivariant Convolutional Networks.” ICML 2016.

---

### Official Review · AnonReviewer1 · 2019-10-23
**Official Blind Review #1**

**Rating:** 3

**Review:**

This paper is introducing a scale-equivariant CNN architecture with joint convolutions over spatial and scale space. Moreover, the authors used decomposable convolutional filters to reduce the number of parameters. Based on Therom 1, it is shown that scale-equivariance is achieved if and only if joint convolutions are conducted over spatial and scale space.

Overall, the contribution seems meaningful but incremental that the method is almost similar to the 'RotDCF: Decomposition of convolutional filters for rotation-equivariant deep network'.

- to verify scale equivariance, visualizing features with tsne would be interesting
- in table 1, what is the reason for doing experiments with and without batchnorm?
- what if the baseline's size is close to the proposed method, how it would be improved?
- it would be better to show how the decomposed filters reduce the total number of parameters.
- also comparing flops would be informative.
- the author's name of 'Locally scale-invariant convolutional neural networks' is wrongly written.

**Experience Assessment:**

I do not know much about this area.

**Review Assessment: Checking Correctness Of Derivations And Theory:**

I did not assess the derivations or theory.

**Review Assessment: Checking Correctness Of Experiments:**

I assessed the sensibility of the experiments.

**Review Assessment: Thoroughness In Paper Reading:**

I made a quick assessment of this paper.

---

> ### Author Response · Authors · 2019-11-15
> **Response to R1**
>
> We would like to thank the reviewer for reading our manuscript and providing valuable comments. Please see below our detailed response, and the manuscript has also been updated accordingly.
>
>
> — Overall, the contribution seems meaningful but incremental that the method is almost similar to the “RotDCF: Decomposition of convolutional filters for rotation-equivariant deep network”.
>
> We would like to thank R1 for pointing out the contribution of our paper. Meanwhile, we would also like to emphasize that our contribution is not just incremental compared to RotDCF: First, the scaling group, unlike SO(2), is non-compact, and the generalization of the group convolution on SO(2) to the scaling group is non-trivial (R3 has kindly pointed this out.) For example, [1] correctly proposed a rotation-equivariant CNN by encoding the rotation information in the phase angle of a vector field; however, its direct generalization to scale-equivariant CNN in [2] is mathematically flawed, for the scaling group has been treated in [2] as a compact cyclic group. Second, the representation stability of the proposed ScDCFNet is also a non-trivial extension of  RotDCFNet —because of the non-compactness of the scaling group, we can only quantify the representation stability in the L_infinity norm instead of the regular L_2 norm (this has been explained in Remark 2.)
>
>
> —To verify scale equivariance, visualizing features with tsne would be interesting.
>
> t-SNE is indeed an interesting tool to visualize the representation of a collection of data. However, our intention is to verify scaling-translation-equivariance on the image level, i.e., roughly speaking, if an input image is spatially rescaled, is its feature map rescaled accordingly (or does equation (4) hold true?) Using t-SNE visualization will not achieve this goal.
>
>
> —In table 1, what is the reason for doing experiments with and without batchnorm?
>
> The reason why we include experiments with and without batch-normalization is that the batch-normalization module needs to be modified for ScDCFNet so that scaling-translation-equivariance still holds true. This has been explained in Appendix B.1 and Section 5 (right before Section 5.1.)
>
>
> — What if the baseline's size is close to the proposed method, how it would be improved?
>
> Table 1 shows the accuracy of our proposed model while changing the model size, i.e., changing the number of low-frequency components (K and K_\alpha) that are retained. In general, as the model size increases (i.e., larger K and K_\alpha), the accuracy of the proposed model is improved. However, as pointed out also in our response to R4, K and K_\alpha cannot be too large because otherwise aliasing would occur. Another way to increase the model size of ScDCFNet is to increase the number of unstructured channel M. In particular, when trained on SMNIST with 2000 images, if we set K = 5 and K_\alpha = 2, while at the same time change M from 16 to 32, the accuracy of ScDCFNet increases from 93.51% to 93.76%, while the model size is now similar to that of the baseline CNN (with an accuracy of 92.60%.)
>
>
> — It would be better to show how the decomposed filters reduce the total number of parameters.
>
> The reason why (and how much) truncating the filter decomposition can reduce the number of parameters is explained in Section 3.2 on page 4 (under “Number of trainable parameters”.)
>
>
> —Also comparing flops would be informative.
>
> The comparison of computational flops is explained in Theorem 2 and proved in Appendix A.2.
>
>
> — The author's name of “Locally scale-invariant convolutional neural networks” is wrongly written.
>
> Thank you for pointing that out. This has been fixed in the revised paper.
>
>
> [1] Diego Marcos et al, “Rotation Equivariant Vector Field Networks.” ICCV 2017.
> [2] Diego Marcos et al, “Scale Equivariance in CNNs with Vector Fields.” arXiv: 1807.11783, 2018.

---

### Official Review · AnonReviewer4 · 2019-11-03
**Official Blind Review #4**

**Rating:** 6

**Review:**

The paper proposes a scale-equivariant Neural Networks model to solve the multi-scale image classification task. The main contributions are the proposed joint convolution across the spatial and scale space, and the decomposed filters to reduce computation cost and improve robustness.

The paper is generally well-written and well placed in the literature. Different from existing work on the scale-equivariance CNNs, they build a more general model that allows different scale information to transfer through different layers. The experiments also suggest the model achieving scale-equivariance and the superior performance of the proposed method compared with several baselines. However, for the novelty part of the proposed method (the separable basis decomposition), it is not convincing enough because the differences between this paper and Cheng et al. (2019) and Weiler et al. (2018b) are not clear.

Questions:
1. The authors claim that the joint convolutions across the space and the scaling group are both "sufficient" and "necessary". Although the experiments show that the proposed method achieves better performance, which might indicate the "sufficiency" of the architecture, however, how should the "necessity" be proved?
2. How should "K" and "K_\alpha" be selected (Table 1)? When data is enough (e.g., 5000), it seems that only some specific values can improve performance.
3. How much can the model size be reduced compared with other multiscale image classification models? And could you provide some concrete comparisons about the computation cost?

Typos:
- 1st line in 2nd paragraph of Introduction: rotation-equivarianc(e)

**Experience Assessment:**

I do not know much about this area.

**Review Assessment: Checking Correctness Of Derivations And Theory:**

I did not assess the derivations or theory.

**Review Assessment: Checking Correctness Of Experiments:**

I assessed the sensibility of the experiments.

**Review Assessment: Thoroughness In Paper Reading:**

I read the paper at least twice and used my best judgement in assessing the paper.

---

> ### Author Response · Authors · 2019-11-15
> **Response to R4**
>
> We would like to thank the reviewer for reading our manuscript and providing valuable comments. Please see below our detailed response, and the manuscript has also been updated accordingly.
>
> — The authors claim that the joint convolutions across the space and the scaling group are both “sufficient” and “necessary”. Although the experiments might indicate the “sufficiency” of the architecture, however, how should the “necessity” be proved?
>
> Our claim in Theorem 1 is that conducting joint-convolutions is both “sufficient” and “necessary” to achieve “scaling-translation-equivariance”, i.e., equation (4) holds true for all l. The proof of “necessity” (as well as “sufficiency”) is detailed in Appendix A.1. The better performance in the experiments on multiscale image classification is a result of the fact that “scaling-translation-equivariance” has been achieved through joint-convolutions.
>
>
> — How should “K” and “K_\alpha” be selected (Table 1)?
>
> The numbers of basis functions, i.e., “K” and “K_\alpha”, cannot be set to too large because otherwise aliasing effects would occur (this is similar to [1].) On the other hand, setting “K” and “K_\alpha” to too small will also limit the expressive power of the network. Our experiments show that setting “K = 8” and “K_\alpha = 3” leads to consistently improved performance while using only a fraction (67%) of the trainable parameters when compared to a regular CNN.
>
>
> — How much can the model size be reduced compared with other multiscale image classification models? Concrete comparisons about the computation cost?
>
> The comparison of the model size and performance between the proposed method and other multiscale image classification models has been added in Table 1. The detailed analysis of the computational cost is stated in Theorem 2 and proved in Appendix A.2. In particular, when basis functions are truncated to “K = 8” and “K_\alpha = 3”, we achieve 80% reduction in computational cost.
>
> [1] Maurice Weiler et al., “Learning Steerable Filters for Rotation Equivariant CNNs.” CVPR 2018.

---

### Public Comment · ~Ivan_Sosnovik1 · 2019-10-16
**On the Related Work**

Thank you for the interesting paper!
While you discuss the papers on scale-equivariance in the Related Work, they are not compared against your method. I propose to conduct the experiments on MNIST-scale with 10k/50k train/test splits as in [1,2] and compare the obtained results with other methods for scale equivariance/invariance in order to better position your paper in the framework of scale-equivariant CNNs.
Additionally, papers [3] and [4] are relevant to the current research.

[1] Marcos D. et al. "Scale equivariance in CNNs with vector fields"
[2] Kanazawa A. et al. "Locally Scale-Invariant Convolutional Neural Networks"
[3] Ghosh R., Gupta A. K. "Scale Steerable Filters for Locally Scale-Invariant Convolutional Neural Networks"
[4] Worrall D. E., Welling M. "Deep Scale-spaces: Equivariance Over Scale"

---

> ### Author Response · Authors · 2019-10-17
> **Thank you for your comment!**
>
> Thank you very much for your interest in our work, and pointing us to the interesting recent development [3] and [4] in the same field. We will include these two papers in our references.
>
> For [3], the authors introduced a very interesting idea of using scale steerable filters, which circumvents the need of interpolation. But the general joint-convolution proposed to impose scale-equivariance in our work has not been studied in [3].
> The scale-space semi-group correlation in [4] does bear a certain resemblance to our joint-convolution. However, the author in [4] did mention that the scale-space correlation only works for discrete semigroups.
>
> As suggested, we have also conducted a preliminary experiment on the MNIST-scale dataset with the 10k/50k train/test splits as in [1,2]. For the case where $K = 8, K_{\alpha} = 2, M = 16$, the error of our proposed network (after six independent trials) is $2.32\pm 0.09$ (compared to $2.75\pm 0.09$ in [2] and $2.44\pm 0.07$ in [1].) Moreover, by taking a truncated filter decomposition, the number of trainable parameters of our network is also smaller than that in [1] and [2]. A full comparison of the results will be updated in Table 1 in the final version of the paper.

---

### Decision · Program_Chairs · 2019-12-19

**Decision:**

Reject

**Comment:**

This paper presents a CNN architecture equivariant to scaling and translation which is realized by the proposed joint convolution across the space and scaling groups. All reviewers find the theorical side of the paper is sound and interesting. Through the discussion based on authors’ rebuttal, one reviewer decided to update the score to Weak Accept, putting this paper on the borderline. However, some concerns still remain. Some reviewers are still not convinced regarding the novelty of the paper, particularly in terms of the difference from (Chen+,2019). Also, they agree that experiments are still very weak and not convincing enough. Overall, as there was no opinion to champion this paper, I’d like to recommend rejection this time.
I encourage authors to polish the experimentations taking in the reviewers’ suggestions.